# Bridging Indexing Structure and Graph Learning: Expressive and Scalable Graph Neural Network via Core-Fringe

## Abstract

Existing message passing-based and transformer-based graph neural networks (GNNs) can not satisfy requirements for learning representative graph embeddings due to restricted receptive fields, redundant message passing, and reliance on fixed aggregations. These methods face scalability and expressivity limitations from intractable exponential growth or quadratic complexity, restricting interaction ranges and information coverage across large graphs. Motivated by the analysis of long-range graph structures, we introduce a novel Graph Neural Network called Core-Fringe Graph Neural Network (CFGNN). Our Core-Fringe structure, drawing inspiration from the graph indexing technique known as Hub Labeling, offers a straightforward and effective approach for learning scalable graph representations while ensuring comprehensive coverage of information. CFGNN leverages this structure to enable selective propagation of relevant embeddings through a carefully designed message function. Theoretical analysis is presented to show the expressivity and scalability of the proposed method. Empirically, CFGNN exceeds standard GNNs on tasks including classification and regression, especially for large, long-range graphs where scalability and coverage matter. Ablation studies further confirm the benefits of our core-fringe based graph neural network, including improved expressivity and scalability.

## 1 Introduction

Graph neural networks (GNNs) provide a powerful framework for learning structural and relational representations of nodes in graphs. These have been extensively used in many application domains to learn node embeddings that capture the structural and relational information in graphs, including social network analysis Leskovec & Mcauley (2012), recommendation systems Fan et al. (2019), citation network analysis Waikhom & Patgiri (2021), and life science Wang et al. (2022b). Two main GNN training methods are commonly used, which are the message passing Wei et al. (2021); Hamilton et al. (2017) and the transformer-based Velickovic et al. (2018); Mialon et al. (2021) approaches. The message passing approach iteratively aggregates messages (features) for each node from its neighbors to update its embedding in order to propagate information across the graph. The transformer-based approach applies the self-attention mechanism from transformers to learn node embeddings based on the contextual information of the entire graph. Both approaches aim to provide accurate representations of all nodes in the graph to support various downstream tasks. The choice of training approaches depends on the graph type and desired properties in the learned embeddings.

To learn the representation of each node by the message passing-based approach, the core idea is to iteratively obtain the information from its neighbors, e.g., $r$-hop neighbors ($r = 3$ for the red dashed circle in Figure 1). By setting $r$ to be the diameter of the graph, this approach can fully obtain both the node and edge information of this graph for each node in order to increase the expressivity of the representation. Although the message passing-based approach can guarantee to cover all information of each node, it necessitates $r$ iterations to learn the representation, with each iteration involving computation of messages for every edge in $E$. which cannot be *scalable* to large (or long-range) graphs with high diameter values. To address this issue, recent research studies Zeng et al. (2020); Ying et al. (2018); Chiang et al. (2019) combine the message passing-based approach with sampling

and clustering techniques to improve scalability. However, these solutions may not fully capture complete information for each node, potentially reducing the *expressivity* of the representation.

Another approach is to use transformers to learn node embeddings, without message passing. The core idea is to apply multi-head self-attention, allowing each node to obtain information from all other nodes in the graph. Unlike message passing, the transformer strategy avoids the iterative process, but it requires $O(|V|^2)$ space to store the self-attention matrix, which limits its scalability to large graphs. Additionally, it does not explicitly model the edge information (graph structure), relying solely on node co-occurrence for learning embeddings, which restricts its expressivity in capturing structural roles and properties of nodes. These issues constrain its performance from explicitly modeling edges between nodes and propagating information in a more structured way for applications that require capturing fine-grained structural node properties.

Hence, we ask a question in this paper. *Can we develop a new GNN structure that provides a highly expressive yet scalable solution?* To provide an affirmative answer to this question. We first propose a new perspective on graph learning problems by formulating the concept of coverage for graph learning tasks

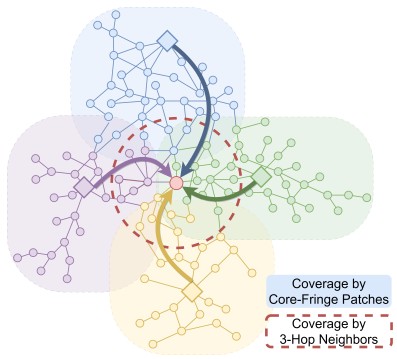

Figure 1: While expanding the number of hops in a GNN poses challenges for the receptive field to encompass all nodes in the graph (red dashed circle), our method overcomes this limitation by achieving complete coverage through the partitioning of the graph into a Core-Fringe structure.

and then discuss the expressivity and scalability issues of existing graph learning methods. Considering the pros and cons of existing methods, we propose a core-fringe structure (cf. Figure 1) that aims to achieve high expressivity and high scalability. The core-fringe structure is constructed using hub labeling Cohen et al. (2003), a graph indexing technique that imparts several crucial properties, e.g., 2-hop cover, shortest path, and index size optimization, for graph learning. A two-stage learning framework is then proposed based on the core-fringe structure. Our approach not only ensures full coverage of every node in the learning process but also offers relatively lower complexity compared with existing approaches due to the two-stage learning framework.

The key contributions of the proposed method can be summarized as follows.

- We introduce a novel graph neural network learning framework based on the core-fringe structure (CFGNN). Our CFGNN is shown to achieve a complete receptive field, ensuring full information coverage for each node in the graph.
- We provide theoretical analysis (including over-squashing analysis and positional-awareness) and empirical results on diverse datasets to demonstrate the expressivity of our approach.
- We present complexity analysis and conduct experiments to showcase the significant improvement in scalability. Remarkably, our CFGNN achieves better or comparable results on a consumer-grade GPU, outperforming state-of-the-art methods that require multiple high-end GPUs.

## 2 ANALYZING GRAPH LEARNING MODELS

### 2.1 INFORMATION COVERAGE

With the growing popularity of GNNs, capturing long-range interactions is crucial for various practical tasks, particularly those involving very large graphs or long chain structures Dwivedi et al. (2022b). For instance, the chemical property of a molecule Ramakrishnan et al. (2014); Gilmer et al. (2017) depends on the combination of atoms situated on opposite sides. To quantify the extent of expressive power in modeling long-range correlations in graphs, we assess the information coverage provided by graph neural networks using the concept called *receptive field* in deep learning, defined as the input size that produces a feature. In a typical Residual Neural Network (ResNet) He et al. (2016), the receptive field at Layer 4 of resnet_v1_101 is approximately $1000 \times 1000$ pixels.

In graph learning, *receptive field* refers to the number of graph nodes contributing to the feature generation process Quan et al. (2019). For example, in a message passing-based GNN, the receptive field encompasses the 2-hop neighbors when the model iterates twice to compute node embeddings. We claim a *receptive field* is *full* for a node $u$ in a graph $G$ (Proposition 1) if the learning process covers all nodes reachable from $u$ in $G$.

We expanded the definition of the prior receptive field to encompass information coverage. Information coverage now includes not just the node feature, but also the feature of the interactions between nodes.

**Proposition 1** (Information Coverage). *The embedding representation of node $u$ is considered to achieve information coverage if it incorporates context information from all of its neighboring nodes that are reachable via any simple path. This concept is referred to as a full information. Mathematically, we define this as follows:*

$$z_u = \tau(V_{[u]}, \mathcal{X}_{[u]}) \tag{1}$$

*where $V_{[u]}$ represents the induced node set that can be reached from node $u$ in graph $G$. The embedding function, denoted as $\tau$, maps each node $v$ in $V_{[u]}$ along with its relevant features $\mathcal{X}_{[u]}$.*

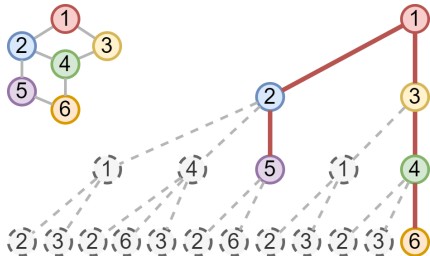

Figure 2: The dash lines indicate all possible paths of length 3 starting from $v_1$, The paths in red achieve information coverage.

Similar to other transformer-based GNNs, the relevant features $\mathcal{X}_{[u]}$ are employed to complement the topological structures as feature enhancements, as seen in prior works such as Ying et al. (2021); Rampášek et al. (2022); Chen et al. (2022a); Dwivedi et al. (2022a). As an example, $\mathcal{X}_{[u]}$ may include a set of simple paths $P_{[u]}$, ensuring that at least one simple path exists from node $u$ to node $v \in V_{[u]}$. As shown in Figure 2, the induced node set $V_{[v_1]}$ encompasses all vertices in the graph since every node is reachable from $v_1$. An example of a simple path within $P_{[v_1]}$ is $v_1 \rightarrow v_3 \rightarrow v_4 \rightarrow v_6$.

## 2.2 Expressivity and Scalability

Existing GNN methods often encounter practical limitations that hinder their effectiveness. As the scale of the graph increases, the computational demands and memory requirements of these methods become impractical Hu et al. (2021). This presents issues with both expressivity and scalability. In general, highly expressive GNN models require complete graph information to achieve a sophisticated understanding of dependencies and relationships. Nevertheless, the scaled-up versions of these models encounter computational challenges when processing information for large graphs. On the other hand, simpler models that are more scalable compromise expressivity, limiting their representational power. We conduct an analysis of two representative graph learning approaches to better illustrate the relationship between expressivity and scalability.

**Message passing-based GNNs.** The representative studies in this category include GCN Kipf & Welling (2017), GraphSage Hamilton et al. (2017), DeeperGCN Li et al. (2018), etc. The detailed formulation of these types of methods is provided in Appendix A.1.2. As shown in Figure 3a, according to Proposition 1, to learn the representation of a node $u$ with *complete* neighborhood information, the model should be able to receive information from other nodes at a distance of $r$, where $r$ indicates the maximum hop distance from $u$ to any nodes in the graph. We should at least stack the layers (the depth of the model) up to $r$, which results in a computation cost of $|E| \cdot r$. Indeed, the message passing-based GNNs face challenges in scaling effectively with increasing values of the receptive field length, $r$.

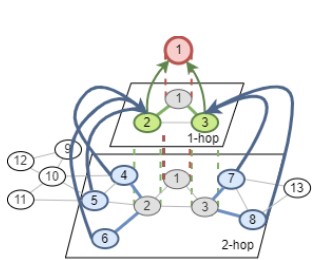 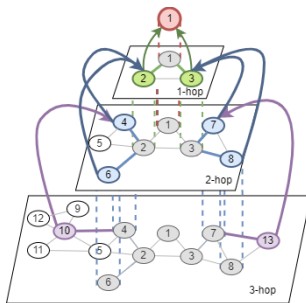

(a) A 2-layer message passing can cover all information in 2-hop neighborhood of the root node in red.

(b) Sampling sacrifices completeness ($v_5$ not sampled) but increases the coverage range (3-hop layer).

Figure 3: An illustration of multi-hop message passing.

To allow GNNs to be used on large graphs, a practical solution is to sample partial messages from the $r$-hop neighbors as shown in Figure 3b, which plays a trade-off between computation resources and information coverage. Although many subgraph sampling techniques, such as neighbor sampling Hamilton et al. (2017), layer sampling Zou et al. (2019), and sub-graph sampling Zeng et al. (2020), have been proposed to tackle the neighbor explosion issue, all these approaches still suffer from certain *expressivity* loss since none of them secures the information coverage (see Figure 3b). If the sampled size of nodes or sub-graphs is too small, some essential structures in the graph may be lost, and there is no assurance that every valuable node and its corresponding path will be considered. Consequently, the topological information may be lost, resulting in a break in the information coverage (Proposition 1).Furthermore, these subgraph sampling techniques still exhibit limited scalability for attaining complete coverage, as the computational complexity increases proportionally with the coverage range in the graph (cf. Figure 3b).

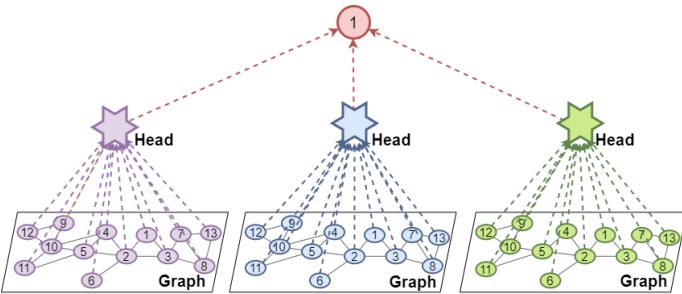

Figure 4: Each attention head in a self-attention transformer model covers all nodes, causing the model to be costly.

**Transformer-based GNNs.** The representative studies in this category include SAN Kreuzer et al. (2021), GraphTrans Wu et al. (2021), Graphormer Rampášek et al. (2022), etc. The detailed formulation of these types of methods is provided in Appendix A.1.3. The self-attention module of a transformer possesses a global receptive field that allows each input token to attend to and process the representation of information at any position; thus, each node in the graph can attend to all other nodes in this model. Nevertheless, the self-attention module overlooks the crucial topological structure between nodes. In other words, the relationship between node pairs is solely captured based on their individual node features. According to Proposition 1, the graph learning should be learned not only from the features but also from how the nodes are connected on a graph in order to supplement more information. Previous studies Ying et al. (2021); Rampášek et al. (2022); Chen et al. (2022a); Dwivedi et al. (2022a) share the same intuition and have attempted to explicitly encode the correlated topological information for each node to supplement the topological structures and positional encoding as a feature enhancement. Graphormer Ying et al. (2021) is one of the transformer-based models that guarantee the information coverage (Proposition 1). It preprocesses and stores all edge encoding that encodes all path features along the shortest path between each pair of nodes.

While transformer-based solutions effectively ensure comprehensive information coverage (as demonstrated in Proposition 1), it is important to acknowledge the well-known scalability challenge

associated with self-attention: the quadratic growth in training time and space complexity $O(|V|^2)$ with the increasing number of graph nodes. Besides, the design of multi-head attention Vaswani et al. (2017) allows the model to simultaneously focus on various aspects of the input sequence, enabling it to capture greater nuance and complexity in the data. However, existing transformers on graph Rampášek et al. (2022); Kreuzer et al. (2021); Wu et al. (2021) do not make optimal use of this property, as all the heads receive input from all the nodes (see Figure 4). Compared to the shorter sequence lengths typically encountered in language processing (usually less than a hundred), the number of nodes in a graph can be significantly larger. Managing such lengthy sequences can pose challenges, particularly in scenarios with limited computational resources. The quadratic complexity associated with self-attention introduces inefficiencies in finding, storing, and modeling large-scale graphs, further exacerbating the computational burden.

**Summary.** To the best of our knowledge, none of the existing methods have successfully addressed both the information coverage (expressivity) and the computational complexity (scalability) challenges simultaneously. This drawback motivates us to develop a novel approach to achieve both objectives. We assert that a robust method should possess the potential for expressive power, enabling capturing long-range interactions (for expressivity) while maintaining reasonable computational complexity for training node embeddings (for scalability). It is crucial for such a method to operate within the limitations of available GPU memory and computational resources, ensuring practical feasibility in real-world applications.

## 3  CORE-FRINGE GRAPH NEURAL NETWORK

### 3.1  CORE-FRINGE STRUCTURE

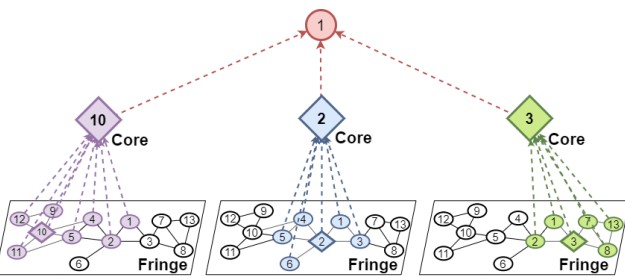

Figure 5: Each core acts as an attention head, and it only focuses on its corresponding fringe.

One approach to alleviate the complexity of multi-head attention is through utilizing sparse attention mechanisms Choromanski et al. (2021). It motivates us to allow different attention heads to focus more selectively on various parts of the graph, as shown in Figure 5. We anticipate that the attention head will prioritize localized information before gradually shifting towards global information rather than immediately allocating equal attention to all nodes. To achieve this division strategy, we employ representative nodes in the graph as *core* nodes and designate their neighboring nodes as the *fringe*. By adopting this approach, we can significantly reduce computational costs, as the attention mechanism only needs to consider a small local neighborhood surrounding each core rather than the entire graph. Their respective core nodes initially collect the fringe messages, ensuring that each core-fringe pair contains partial graph information.

In the context of graph analysis, a *core* node should possess significance within the graph structure. Node properties such as degree can be used to identify the graph core. For instance, celebrities with many followers can be treated as cores in a social network, while in a citation network, a groundbreaking paper can be identified by its citation degree. Complex networks, such as social and biological networks, exhibit small-world properties, making the core-fringe structure a prevalent property of these. Note that a core may also serve as the fringe of another core.

Some approaches in graph pooling Chiang et al. (2019); Cai et al. (2021); Ying et al. (2018); Yuan & Ji (2020); Murphy et al. (2019); Mesquita et al. (2020) tackle the scalability issue by formulating it as a *clustering* problem or a *graph cut* problem, extending the concept of local patches (a subsection of an image) in regular pixel grids to graphs. However, while these methods can achieve high receptive

fields through a hierarchical approach, they may inadvertently destroy the topological structure, which can impact the expressivity power of the model. An evident concern is that incorrect *cutting* of an edge in the coarser graph may lead to the loss of certain connectivity information, such as shortest paths. We will further analyze the importance of the topological information in Appendix A.6.

## 3.2 BRIDGING GRAPH LEARNING AND INDEXING

In this work, we propose to leverage indexing techniques from graph query answering to overcome the aforementioned limitations. Graph indexing techniques, in general, are designed to efficiently answer queries such as the shortest path by utilizing specialized data structures. Graph query answering and graph learning share a common objective of accomplishing tasks with minimal resource utilization. While graph query answering focuses on ensuring query correctness, graph learning aims to achieve learning efficiency, such as information coverage.

Specifically, we construct the core-fringe structure using the Hub Labeling (HL) technique Cohen et al. (2003). HL is a graph indexing technique that optimizes shortest path-finding algorithms, trading space for time compared to conventional Dijkstra's algorithm Dijkstra (1959). In the following, we will outline our approach for utilizing HL techniques to construct the core-fringe structure. We will then discuss how the constructed core-fringe satisfies the desired properties.

**Definition 3.1** (Hub Label). *For each node $v \in V$, we define the hub label of a node $L(v)$ as a set of pairs $(h, dist)$, where $dist$ is the distance from $v$ to the hub vertex $h$.*

The fundamental idea behind hub labeling is to pre-compute and store the information about the distances between each vertex and a small set of hubs, nodes with a high degree and many connections to other vertices in the graph. The distance information is stored in a data structure, namely a hub label. To compute the result of the shortest path query from source $u$ to destination $v$, a sort-merge join is performed between its hub labels $L(u)$ and $L(v)$. The purpose of the sort-merge join is to find a common hub in the shortest path from $u$ to $v$ that is labeled by both $u$ and $v$. The construction methodology is outlined in Appendix A.3.

**Property 1** (2-hop cover). *For any pair of reachable nodes $u, v \in V$ of $G(V, E)$, there exists at least one common hub $h \in SP_{u \to v}$ in both label sets, $L(u)$ and $L(v)$, such that the shortest path $SP_{u \to v}$ is the result of merging $SP_{u \to h}$ and $SP_{h \to v}$.*

$$dist(SP_{u \to v}) = \min_{h \in L(u) \cap L(v)} \{dist(SP_{u \to h}) + dist(SP_{h \to v})\} \tag{2}$$

To ensure the correctness of the shortest path finding, the hub label of each node must satisfy the 2-hop cover property (see Property 1). An optimization goal in hub label construction is typically to minimize the number of labels, i.e., $\min \sum_{u \in V} |L(v)|$. In Babenko et al. (2015), it shows that finding a hub labeling with the minimum total label size while maintaining the 2-hop cover property is a formulation of the NP-hard weighted set-cover problem Abraham et al. (2012). Similar to solving set-cover problems, many greedy heuristic algorithms have been shown to provide good approximations in practice Abraham et al. (2012); Akiba et al. (2013); Li et al. (2017).

**HL-based core-fringe structure.** In order to understand the concept of the core set and the fringe set, let us explore an example using a node called $v_1$ depicted in Figure 5. In this case, $v_1$ has three cores, namely $v_2$, $v_3$, and $v_{10}$. The hub labels of each core in the core set are used to create the fringe set. For instance, the nodes highlighted as purple color are the fringe set of core $v_{10}$.

## 3.3 CORE-FRINGE BASED GRAPH NEURAL NETWORK

**Matrix representation for graph learning.** To provide a clear contrast between our model and the other frameworks for graph learning, we present graph learning in a simplified matrix multiplication format. The $r$-layer message passing methods on node neighbors can be written as the following expression.

$$Z = \rho(\mathbb{A}_N...(\mathbb{A}_N(\mathbb{A}_N X W_1) W_2)...W_r) \tag{3}$$

where $\mathbb{A}_N$ indicates the neighbor affinity matrix of neighbors $N$. Obviously, the message passing-based methods involve $r$ iterations of matrix multiplication, enabling a node to receive messages from $r$-hop neighbors (full receptive field).

The transformer-based methods only require one step since the self-attention affinity matrix $\mathbb{A}_V$ encompasses all pairs of nodes.

$$Z = \rho(\mathbb{A}_V \, XW) \tag{4}$$

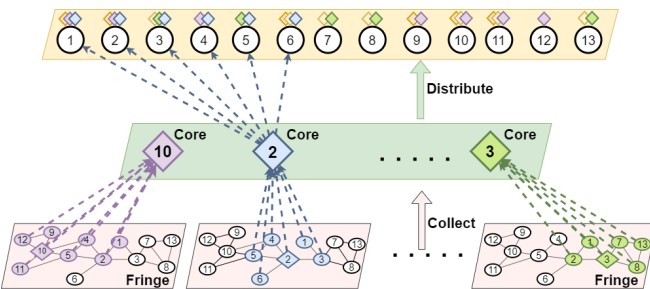

Figure 6: Cores are considered transit hubs of the message passing. Each of these cores first collects the messages from fringes and updates its representation, and then distributes its updated representation to its fringes.

In contrast to these two classic methods, the core-fringe structure allows us to achieve Proposition 1 with only a two-stage model as illustrated in Figure 6. The matrix multiplication format of the process can be presented as follows.

$$Z = \rho(\overbrace{\mathbb{A}_L^T(\underbrace{\mathbb{A}_L XW_1}_{\mathcal{F} \to \mathcal{C}})W_2}^{\mathcal{C} \to \mathcal{F}}) \tag{5}$$

where $\mathbb{A}_L$ indicates the affinity matrix of hub labels. For clarity, the mathematical equations of the two-stage process are shown in message passing fashion as follows.

$$
\begin{array}{clcl}
 & \text{Fringe} \to \text{Core} & & \text{Core} \to \text{Fringe} \\
\text{Message:} & m_{f \to c}^k = \phi(\boldsymbol{z}_c^{k-1}, \boldsymbol{z}_f^{k-1}), \forall f \in \mathcal{F}(c) & m_{c \to f}^k = \phi(\boldsymbol{z}_f^{k-1}, \boldsymbol{z}_c^{k-1}), \forall c \in \mathcal{C}(f) \\
\text{Update:} & \boldsymbol{z}_c^k = \rho(\boldsymbol{z}_c^{k-1}, \oplus(\{m_{f \to c}^k\})) & \boldsymbol{z}_f^k = \rho(\boldsymbol{z}_f^{k-1}, \oplus(\{m_{c \to f}^k\}))
\end{array} \tag{6}
$$

**Collect and Distribute.** Our Core-Fringe based GNN (CFGNN) involves two stages. The first stage collects messages from fringes to learn the core embeddings, i.e., $\mathcal{F} \to \mathcal{C}$. The second stage distributes the messages from cores back to all fringes, i.e., $\mathcal{C} \to \mathcal{F}$. As depicted in Equation 5, these two stages are a pair of mirroring processes. The first stage *collects* information from fringes using $\mathbb{A}_L$, while the second stage *distributes* the aggregated information to the targeted nodes based on the transposed matrix $\mathbb{A}_L^T$. An algorithm outlining the framework is provided in Appendix A.4.

In the first stage, messages are collected in a message passing fashion. Cores collect and aggregate messages from all their fringes to update their embeddings (lower part of Figure 6). This is achieved by computing a message vector for each fringe node via the message function and then aggregating them using a differentiable and permutation-invariant function Equation 6).

In the second stage, the representation of the cores contains all the information from their fringes (upper part of Figure 6). It is worth noting that the union of the fringe set is equivalent to the target node set due to the 2-hop cover property (Property 1). In the subsequent stage, a *mirror* process is used to update the representation of the fringes, which distributes the messages from the cores back to the fringes. Each fringe will receive messages from its core set and update its representation, similar to how the previous stage updated the cores Equation 6).

### 3.4 THEORETICAL ANALYSIS AND COMPUTATIONAL COMPLEXITY OF CFGNN

**Lemma 1.** *HL-based core-fringe fulfills Proposition 1.*

CFGNN, which integrates hub labeling into the field of graph learning, provides a powerful methodology for achieving comprehensive information coverage. This approach effectively captures long-range dependencies while ensuring efficiency. For a detailed proof, please refer to Appendix A.2.

**Lemma 2.** *The over-squashing phenomenon of CFGNN is lower than that of other message passing-based GNNs.*

By mitigating the issue of excessive compression, CFGNN guarantees the preservation of the graph's integrity and pertinent information. This allows for the propagation of node information across longer distance, hence enabling the capture of more complex structures. The proof of this lemma is shown in Appendix A.2. Moreover, The core-fringe structure allows for efficient information propagation with significantly fewer iterations compared to message passing based methods and uses a much sparser matrix compared to transformer based methods. A detail complexity analysis is provided in Appendix A.7 and the complexity is shown in Table 4.

## 4 EXPERIMENTS

This section evaluates the effectiveness of the CFGNN for graph tasks, and our implementation is developed based on the GraphGym You et al. (2020) module of Pytorch-Geometric Fey & Lenssen (2019). A computation resource is a single machine with an NVIDIA RTX3090 GPU with 24GB GPU memory and an AMD Ryzen Threadripper 3960X CPU with 24 cores and 64GB RAM. Besides, our source codes can be found in https://anonymous.4open.science/r/CFGNN-5FFC/. Additional information regarding the experimental settings, bechmark and baselines can be found in Appendix A.8 and A.9.

For certain methods, we report the best performance from the original papers as some of the hyper-parameter configurations in previous works are not publicly available. In the tables, we highlight the methods with a star * to indicate that the results are from the original papers. We also include information about the hardware used in our experiments, including the GPU type and the available GPU memory, to provide a reference for the readers.

Table 1: GNN Benchmark, the result of CFGNN is the mean standard deviation of 5 runs

| Method | Hardware | ZINC | PATTERN | CLUSTER |
|---|---|---|---|---|
| | | MAE ↓ | Accuracy ↑ | Accuracy ↑ |
| *GCN | 1 × Tesla V100, 32GB | 0.367 ± 0.011 | 71.892 ± 0.334 | 68.498 ± 0.976 |
| *GAT | 1 × Tesla V100, 32GB | 0.384 ± 0.007 | 78.271 ± 0.186 | 70.587 ± 0.447 |
| *GIN | 1 × Tesla V100, 32GB | 0.526 ± 0.051 | 85.387 ± 0.136 | 64.716 ± 1.553 |
| *PAN | 1 × Tesla V100, 32GB | 0.188 ± 0.004 | - | - |
| DiffPool-GCN | 1 × RTX 3090, 24GB | 0.324 ± 0.073 | - | - |
| *SAN | 1 × Tesla V100, 32GB | 0.139 ± 0.006 | 86.581 ± 0.037 | 76.691 ± 0.650 |
| *Graphormer | 8 × Tesla V100, 32GB | 0.122 ± 0.006 | - | - |
| Graphormer-small | 1 × RTX 3090, 24GB | 0.1721±0.009 | 84.312 ± 0.064 | 77.831 ± 0.773 |
| *EXPHORMER | 1 × Tesla A100, 40GB | - | *86.740±0.015* | *78.07 ± 0.037* |
| *GraphGPS | 1 × Tesla A100, 40GB | **0.070 ± 0.004** | 86.685 ± 0.059 | 78.016 ± 0.180 |
| CFGNN | 1 × RTX 3090, 24GB | *0.113 ± 0.006* | **86.821 ± 0.026** | **78.863 ± 0.032** |

Table 2: Open Graph Benchmark, the result of CFGNN is the mean standard deviation of 5 runs.

| Method | ogbg-molhiv | ogbg-molpcba | ogbg-ppa | ogbg-code2 |
|---|---|---|---|---|
| | AUROC ↑ | Avg. Precision ↑ | Accuracy ↑ | F1 score ↑ |
| *GCN | 0.7585 ± 0.0061 | 0.2318 ± 0.0032 | 0.6903 ± 0.0068 | 0.1585 ± 0.0021 |
| *GAT | 0.7612 ± 0.0031 | 0.2172 ± 0.0082 | 0.6843 ± 0.0093 | 0.1570 ± 0.0014 |
| *GIN | 0.7707 ± 0.0149 | 0.2703 ± 0.0023 | 0.7037 ± 0.0107 | 0.1581 ± 0.0026 |
| *PAN | 0.7742 ± 0.0091 | 0.2616 ± 0.0057 | 0.7617 ± 0.0116 | 0.1638 ± 0.0220 |
| DiffPool-GCN | 0.7664 ± 0.0088 | 0.2442 ± 0.0075 | 0.7249 ± 0.0084 | 0.1441 ± 0.0035 |
| *SAN | 0.7785 ± 0.2470 | 0.2765 ± 0.0042 | - | - |
| *GraphTrans | - | 0.2761 ± 0.0029 | - | 0.1830 ± 0.0024 |
| *Graphormer | **0.8051 ± 0.5300** | **0.3139 ± 0.3200** | - | - |
| Graphormer-small | 0.7691 ± 0.0176 | 0.2672 ± 0.0132 | - | - |
| *GraphGPS | 0.7880 ± 0.0101 | 0.2907 ± 0.0028 | **0.8015 ± 0.0033** | **0.1894 ± 0.0024** |
| CFGNN | *0.7970 ± 0.0462* | *0.2952 ± 0.0041* | *0.7881 ± 0.0053* | *0.1840 ± 0.0017* |

### 4.1 EXPERIMENTAL ANALYSIS

Table 1 compares the performance of graph neural networks on three datasets, ZINC, PATTERN, and CLUSTER. The evaluation metrics are mean absolute error (MAE) for ZINC and accuracy for PATTERN and CLUSTER. The proposed CFGNN model achieves a low Mean Absolute Error (MAE) of 0.113 on the ZINC dataset, surpassing all baseline methods except GraphGPS, which

Table 3: Long Range Graphs, the result of CFGNN is the mean standard deviation of 5 runs

| Method | PascalVOC-SP | COCO-SP | Peptides-func | Peptides-struct |
|---|---|---|---|---|
| | F1 score ↑ | F1 score ↑ | AP ↑ | MAE ↓ |
| *GCN | 0.1268 ± 0.0060 | 0.0841 ± 0.0010 | 0.5930 ± 0.0023 | 0.3496 ± 0.0013 |
| *GatedGCN | 0.2873 ± 0.0219 | 0.2641 ± 0.0045 | 0.5864 ± 0.0077 | 0.3420 ± 0.0013 |
| *SAN | 0.3230 ± 0.0039 | 0.2592 ± 0.0158 | 0.6384 ± 0.0121 | 0.2683 ± 0.0043 |
| *Transformer+LapPE | 0.2694 ± 0.0098 | 0.2618 ± 0.0031 | 0.6326 ± 0.0126 | 0.2529 ± 0.0016 |
| *GraphGPS | 0.3748 ± 0.0109 | *0.3412 ± 0.0044* | *0.6535 ± 0.0041* | 0.2500 ± 0.0005 |
| *EXPHORMER | **0.3975 ± 0.0037** | **0.3455 ± 0.0009** | 0.6527 ± 0.0043 | *0.2481 ± 0.0007* |
| CFGNN | *0.3847 ± 0.0273* | 0.2810 ± 0.0095 | **0.6581 ± 0.0047** | **0.2477± 0.0059** |

achieves a slightly better MAE of 0.070. For PATTERN and CLUSTER, CFGNN achieves the highest accuracy of 86.821% and 78.863%, respectively, surpassing all models, including GraphGPS and EXPHORMER. Compared to more resource-intensive models, such as Graphormer, which runs on multiple high-end GPUs, CFGNN efficiently achieves better or comparable results on a single consumer-grade GPU. This highlights the benefits of the proposed core-fringe framework and validates the ability to learn graph representations with high coverage and scalability.

Furthermore, Table 2 presents a comparison of GNN models on four Open Graph Benchmark datasets, ogbg-molhiv, ogbg-molpcba, ogbg-ppa, and ogbg-code2. On ogbg-molhiv, Graphormer achieves the best AUROC of 0.8051. However, the proposed CFGNN obtains a competitive AUROC of 0.7970, surpassing all baseline GNNs, including Graphormer-small, a method we ran locally to demonstrate the performance of Graphormer under limited available resources. For ogbg-molpcba, Graphormer again shows the top average precision of 0.3139, while CFGNN attains a close second at 0.2952, outperforming other models. On ogbg-ppa, CFGNN obtains an accuracy of 0.7881, surpassing strong baselines like GIN and PAN. Finally, for ogbg-code2, GraphGPS has the highest F1 score of 0.1894, but CFGNN achieves a very close 0.1840, significantly exceeding GNN baselines and GraphTrans. It is worth noting that the resources required by Graphormer on larger datasets remain unknown, even though it shows top performance on chemical datasets. The scalability and resource efficiency of CFGNN is advantageous in handling larger and more complex graphs.

We also analyze the performance over long-range graph benchmarks shown in Table 3. Again, CFGNN outperforms or remains competitive with state-of-the-art techniques on these long-range graph tasks, showcasing its ability to model complex dependencies in graphs effectively. CFGNN provides an effective practical framework for long-range graph learning. On PascalVOC-SP, the proposed CFGNN achieves a F1 score of 0.3847, significantly outperforming prior works, including SAN, Transformer + LapPE, and GraphGPS. For COCO-SP, EXPHORMER shows the best F1 of 0.3412, but CFGNN obtains a competitive 0.2810, exceeding other methods. For Peptides-func, CFGNN has the top AP of 0.6581, while GraphGPS achieves a close second at 0.6535, surpassing SAN and Transformer baselines. Finally, on Peptides-struct, CFGNN attains the lowest MAE of 0.2477. Ablation study results can be found in Appendix A.10 to A.11.

## 5 CONCLUSION

In this work, we introduce a novel graph learning framework called CFGNN, based on the core-fringe graph structure. Our results demonstrate that CFGNN represents an advancement in the field of graph neural networks, especially beneficial for tasks involving graph data management and analytics. In particular, CFGNN addresses the scalability and expressivity challenges of graph neural networks by introducing an efficient message passing framework for graph learning that employs the hub-labeling method. By attaining full coverage in message passing, CFGNN enables more expressive graph representations, resulting in more potent and accurate predictions. The theoretical and empirical analysis of the proposed model demonstrates its ability to outperform other state-of-the-art models, indicating its potential as a fundamental solution to the fundamental problem of scalability and expressivity in graph neural networks. CFGNN has a wide range of potential applications, and its ability to manage large-scale graphs efficiently makes it a promising solution for various real-world problems. It could be used, for instance, to predict protein interactions in bioinformatics or social interactions in social networks. In the future, we plan to investigate how the structure driven GNN performs in configurations with large memory, enabling training of large-scale models.

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
