# A  APPENDIX

## A.1  PRELIMINARIES

### A.1.1  GRAPH ANALYTICS AND GRAPH LEARNING

**Graph analytics.** Graph analytics aim to capture complex relationships between nodes and their features, which are particularly important in applications such as recommendation systems, social network analysis, and bioinformatics. Node embeddings $\boldsymbol{Z}$ are a crucial aspect of graph analytics. They consist of low-dimensional vector representations of each node in a graph $G = (V, E)$, denoted by $\boldsymbol{z}_u \in \mathbb{R}^h$ for node $u$. Node embeddings encode both the features of a node and the structure of its local graph neighborhood, allowing for more efficient and accurate downstream tasks.

**Graph learning.** Graph Neural Networks (GNNs) have become a widely used approach for learning node embeddings in graph-structured data. Recently, two primary methods for learning embeddings have been using GNNs. The first method is message passing, where information is iteratively aggregated from neighbors to produce an embedding for each node. The second method involves applying a specialized encoder architecture, such as a transformer-based model, to learn embeddings directly from the graph-structured data.

### A.1.2  MESSAGE PASSING-BASED GNNS

Message passing Hamilton et al. (2017) generalizes the convolutional operator of convolutional neural networks (CNNs) to support graph-structured data. The primary goal of message passing is to capture the homophily and structural equivalence of nodes in a graph. During each message passing step, a node $u$ aggregates messages from all of its neighbors $v \in N(u)$. Each message is computed by a differentiable and learnable linear transformation $\phi(\cdot)$.

The node embedding representation $\boldsymbol{Z}$ can be learned by an iterative process. Initially, the embedding of a node $u$ is set to its feature vector, i.e., $\boldsymbol{z}_u^0 = x_u$. At $k$-th iteration, the message $m_e^k$ of edge $e_{v,u}$ is derived from $\boldsymbol{z}_u^{k-1}$ and $\boldsymbol{z}_v^{k-1}$ (the embedding of nodes $u$ and $v$ at $(k-1)$-th iteration) and the features of edge $e_{v,u}$ by the transform function $\phi(\cdot)$. For all received messages $m_e^k$ (from incoming edges) at node $u$, an aggregator $\oplus(\cdot)$, usually a differentiable and permutation invariant function (e.g., Sum, Mean, Max), is then applied to compress the received messages.

Finally, the embedding $\boldsymbol{z}_u^k$ is then updated with the embedding $\boldsymbol{z}_u^{k-1}$ (at the last iteration) and the compressed message $\oplus(\cdot)$ by an update function $\rho$. For clarity, the mathematical equations of the iterative process are shown as follows.

$$\text{Message:} \qquad m_e^k = \phi(\boldsymbol{z}_u^{k-1}, \boldsymbol{z}_v^{k-1}, e_{v,u}), \forall e_{v,u} \in E \qquad (7)$$

$$\text{Update:} \qquad \boldsymbol{z}_u^k = \rho(\boldsymbol{z}_u^{k-1}, \oplus(\{m_e^k | \forall e_{v,u} \in E\})) \qquad (8)$$

### A.1.3  TRANSFORMER-BASED GNNS

*Transformer* Vaswani et al. (2017) is another typical embedding technique promising in addressing long-range interactions in graph data. The transformer architecture is composed of multiple transformer layers, each consisting of two components: a self-attention module and a position-wise feed-forward network. The self-attention module in the context of graphs can be formulated in a message passing style, as described in previous work Velickovic et al. (2018). In this formulation, the messages are not only passed between neighboring nodes, but also between all nodes in the graph (Equation 11 and Equation 12), as self-attention allows for full connectivity. To capture the correlations among all node pairs, an attention coefficient $\alpha_{v,u}$ is defined to substitute the edge feature $e_{v,u}$ when computing the message (Equation 10). The attention coefficient between every pair of nodes is computed by embedding a learnable softmax attention score $s$ produced by a multi-layer perceptron (Equation 9).

Attention score: 
$$s_{v,u} = MLP(\boldsymbol{z}_v, \boldsymbol{z}_u) \tag{9}$$

Coefficient: 
$$\alpha_{v,u} = \frac{exp(s_{v,u})}{\sum_{k \in V} exp(s_{k,u})} \tag{10}$$

Message: 
$$m_{v \to u}^k = \phi(\boldsymbol{z}_v^{k-1}, \boldsymbol{z}_u^{k-1}, \alpha_{v,u}), \forall v \in V \tag{11}$$

Update: 
$$\boldsymbol{z}_u^k = \rho(\boldsymbol{z}_u^{k-1}, \oplus(\{m_{v \to u}^k | \forall v \in V\})) \tag{12}$$

The self-attention mechanism in a transformer empowers each token to incorporate information from any position, offering significant expressive capability for encoding intricate relationships within a sequence. In the context of graph neural networks, this mechanism allows each node in a graph to be considered as an input token during the application of the transformer layer. As a result, every node can attentively access information from all other nodes, enabling adaptive modeling of node-to-node correlations and fostering a comprehensive understanding of the graph structure.

## A.2 CORE-FRINGE-BASED GNN ANALYSIS

**Information coverage.** HL-based Core-Fringe ensures that every shortest path between any pair of nodes must pass through at least one common hub node such that full receptive field is secured. In other words, the hub nodes act as bridges connecting multiple pairs of nodes in the graph, making them strong candidates for good cores. Here, we formally show that HL-based core-fringe can achieve comprehensive information coverage.

**Lemma.** *HL-based core-fringe fulfills Proposition 1.*

*Proof.* By selecting any hub labeling (e.g., Pruned Landmark Labeling (PLL) Akiba et al. (2013)) that satisfies the 2-hop cover property as the core-fringe structure, the union of the fringe set $\mathcal{F}$ of $v$'s core set $\mathcal{C}$ is equivalent to the complete node set $V$ due to the 2-hop cover property (Property 1). Mathematically,

$$V \equiv \mathcal{F} = \cup_{c \in \mathcal{C}(v)} \mathcal{F}(c) \tag{13}$$

This secures that the HL-based core-fringe structure is reachable to all nodes. Besides, the core-fringe structure can identify at least one shortest path between any pair of nodes, thus representing their relationship, as stated in Property 1. □

**Scalability.** In message passing-based GNNs, each node collects messages from its one-hop neighbors $N$, and the affiliation of every node is represented by a matrix $\mathbb{A}_N$ which is the adjacency matrix provided by the graph data. To achieve information coverage (Proposition 1), this process is repeated $r$ times, where $r$ indicates the length of the full receptive field. On the other hand, transformer-based GNNs collect messages from all nodes in one step, and the affiliation is represented by a full matrix $\mathbb{A}_V$. Given the core-fringe structure, each node $u$ collects messages from its hub labels $L(u)$, and the affiliation of every node is represented by a matrix $\mathbb{A}_L$, where $\mathbb{A}_L(u, v) = 1, if v \in L(u)$. According to Lemma 1, achieving full information coverage requires running the message passing exactly two times, as supported by the 2-hop cover property.

The sparsity between the matrices of message passing, core-fringe, and transformer can be generally summarized as follows: $|\mathbb{A}_N| \leq |\mathbb{A}_L| \leq |\mathbb{A}_V|$, with $\mathbb{A}_V$ being the superset of the other two matrices (cf. Equation 13). It is crucial to emphasize that the computation in the learning process is sensitive to the sparsity since we utilize sparse matrix calculations in the learning framework. Interestingly, in our experimental studies (cf. Table 5), the sparsity of $\mathbb{A}_L$ in "COCO-SP" is even better than that of $\mathbb{A}_N$, showcasing the core-fringe structure's advantage. Furthermore, the numbers of iterations for these three approaches are as follows: $r$ (message passing) $> 2$ (core-fringe) $> 1$ (transformer).

Obviously, the core-fringe structure excels when the average size of the hub label set $\ell$ is small. As reported in numerous studies Li et al. (2017); Akiba et al. (2013), the label size of a node is much

smaller than $|V|$ in practice. For instance, there are typically hundreds of labels per node, even in large-scale graphs containing millions of nodes Akiba et al. (2013).

**Expressivity.** In addition to ensuring information coverage, our core-fringe structure empowers the learning process to concentrate on highly informative features, facilitating effective learning while reducing redundancy in the information. To showcase the expressive capability of the core-fringe approach, we conduct a thorough analysis of over-squashing, a concept extensively explored in graph learning studies Topping et al. (2022); Rampášek et al. (2022); Chen et al. (2022b).

The distortion of message passing from one node to another node is caused by the repetition of massive involving nodes from a long range message passing progress. This phenomenon is named as over-squashing Topping et al. (2022). To measure the degree of distortion, the Jacobian method is proposed as a formal way to evaluate the over-squashing phenomenon Topping et al. (2022). The main conclusion of the Jacobian method is that, as the distance between two nodes increases, the issues of distortion and over-squashing become more pronounced. One approach to mitigate over-squashing is to connect all nodes in the graph. However, this would result in the loss of topological information, which is crucial for message passing in classification tasks, especially for prediction tasks that heavily rely on remote interactions. Achieving a balance between reducing over-squashing and preserving topological information presents a significant challenge in graph representation learning.

**Lemma.** *The over-squashing phenomenon of CFGNN is lower than that of other message passing-based GNNs.*

*Proof.* Under the core-fringe structure and Equation 6, based on the Property 1, for a node pair $u$ and $v$, there always exists a common core node $c$. The message passing progress from $v$ to $u$ is as follows.

$$z_v^k = \rho\left(z_v^{k-1}, \oplus\left(\{\phi\left(z_v^{k-1}, \rho\left(z_c^{k-2}, \oplus\left(\{\phi\left(z_c^{k-2}, z_u^{k-2}\right)\}\right)\right)\right)\}\right)\right)$$

According to Topping et al. (2022), the level of message distortion can be estimated by partial derivative. Without loss of generality, in order to simplify the equations, we assume that node features and hidden representations are scalar.

$$
\begin{aligned}
\frac{\partial \boldsymbol{z}_u^{(2)}}{\partial \boldsymbol{x}_v} =& \partial_1 \rho \partial_{\boldsymbol{x}_v} \boldsymbol{z}_{j_2}^{(1)} + \partial_2 \rho \oplus \\
& \left\{\partial_1 \phi\left(\partial_{\boldsymbol{x}_v} \boldsymbol{z}_{f_1}^{(1)}, \rho\left(\partial_{\boldsymbol{x}_v} \boldsymbol{z}_c^{(0)}, \oplus\left(\{\phi\left(\partial_{\boldsymbol{x}_v} \boldsymbol{z}_c^{(0)}, \partial_{\boldsymbol{x}_v} \boldsymbol{z}_{f_0}^{(0)}\right)\}\right)\right)\right) \right. \\
& \left. + \partial_2 \phi\left(\partial_{\boldsymbol{x}_v} \boldsymbol{z}_{f_1}^1, \rho\left(\partial_{\boldsymbol{x}_v} \boldsymbol{z}_c^{(0)}, \oplus\left(\{\phi\left(\partial_{\boldsymbol{x}_v} \boldsymbol{z}_c^{(0)}, \partial_{\boldsymbol{x}_v} \boldsymbol{z}_{f_0}^{(0)}\right)\}\right)\right)\right)\right\}
\end{aligned}
\tag{14}
$$

As a reference, we also show the message distortion estimation for the typical message passing methods. First, we combine Equation 7 and Equation 8 as follows.

$$\boldsymbol{z}_u^{(k)} = \rho\left(\boldsymbol{z}_u^{(k-1)}, \oplus(\phi(\boldsymbol{z}_u^{(k-1)}, \boldsymbol{z}_v^{(k-1)}, e_{v,u}))\right)$$

The **Jacobian** measures how a change of the input feature $\boldsymbol{x}_v$ affects the node output $\boldsymbol{z}_u^{(k)}$. We can compute the influence of $\boldsymbol{x}_v$ on $\boldsymbol{z}_u^{(k)}$ as follows.

$$\frac{\partial \boldsymbol{z}_u^{(k)}}{\partial \boldsymbol{x}_v} = \partial_1 \rho \partial_{\boldsymbol{x}_v} \boldsymbol{z}_{j_{k-1}}^{(k-1)} + \partial_2 \rho \oplus (\partial_2 \phi \partial_{\boldsymbol{x}_v}(\boldsymbol{z}_u^{(k-1)}, \boldsymbol{z}_v^{(k-1)}, e_{v,j_{k-1}}))$$

where $k$ is the distance between nodes $j$ and $u$, $j_k$ is the node with distance $k$ to $v$.

By expanding the layers recursively, we can rewrite the equation for the next expansion as follows.

$$
\begin{aligned}
\frac{\partial \boldsymbol{z}_u^{(k)}}{\partial \boldsymbol{x}_v} =& \partial_1 \rho \partial_{\boldsymbol{x}_v} \boldsymbol{z}_{j_k}^{(k-1)} + \partial_2 \rho \oplus \left(\partial_2 \phi \partial_{\boldsymbol{x}_{j_{k-1}}}(\boldsymbol{z}_{j_{k-1}}^{(k-1)}, \boldsymbol{z}_v^{(k-1)}, e_{v,j_{k-1}})\right) \\
=& \partial_1 \rho \partial_{\boldsymbol{x}_v} \rho\left(\boldsymbol{z}_{j_{k-1}}^{(k-2)}, \oplus(\phi(\boldsymbol{z}_{j_{k-1}}^{(k-2)}, \boldsymbol{z}_{j_{k-1}}^{(k-2)}, e_{v,j_{k-1}}))\right) \\
& + \partial_2 \rho \oplus \left(\partial_2 \phi \partial_{\boldsymbol{x}_v} \rho\left(\boldsymbol{z}_{j_{k-1}}^{(k-2)}, \oplus(\phi(\boldsymbol{z}_{j_{k-2}}^{(k-2)}, \boldsymbol{z}_v^{(k-2)}, e_{j_{k-1},j_{k-2}}))\right)\right)
\end{aligned}
\tag{15}
$$

The message passing steps of CFGNN between two nodes are always fixed at 2 (two hops), which is much shorter than that of traditional message passing-based GNNs. This proximity between nodes results in a significantly lower distortion value compared to the nodes based on other message passing methods. This is also evident from the distortion values calculated by Equation 14, which are much lower than those obtained from the message passing methods using Equation 15.

□

### A.3 CORE-FRINGE CONSTRUCTION

We discuss a widely used canonical HL algorithm Abraham et al. (2012), called Pruned Landmark Labeling (PLL) Akiba et al. (2013), for constructing the core-fringe structure, which maintains an order for all nodes $\mathcal{O}$ (e.g., using the degree value of each node as an order) so that the label set $L(v)$ of each node $v$ only contains higher-order nodes.

In order to construct PLL, this algorithm initially starts from the highest-order node and iteratively (1) propagates the distance information and (2) assign the label from the higher-order node $u$ to other lower-order nodes $v$. During the propagation of node $u$, the process is stopped at node $v$ if the existing label sets $L(u) \cap L(v)$ are already sufficient for computing the shortest path between nodes $u$ and $v$ (cf. Definition A.1). PLL has shown remarkable performance by generating only 250 labels per node in a social network graph (Flickr) of 2.3 million nodes and 33 million edges Akiba et al. (2013).

**Definition A.1** (Pruning Condition). *A hub $h$ is unnecessary to be added in $L(u)$ if there exists a prior hub $p$ that co-exists in $L(h)$ and $L(u)$. Thus, the minimum path distance from $h$ to $u$ can already be determined by merging $dist(SP_{h \to p})$ and $dist(SP_{p \to u})$.*

$$dist(SP_{h \to u}) \geq dist(SP_{h \to p}) + dist(SP_{p \to u})$$

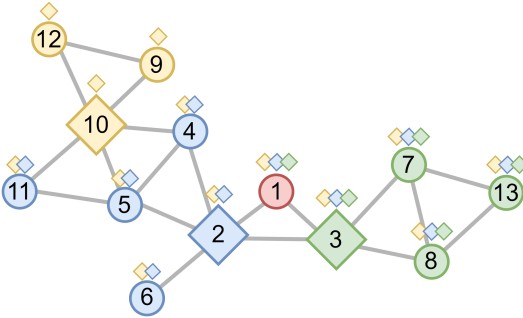

Figure 7: $L(v_1)$ has three labels from $v_{10}$, $v_2$, and $v_3$. To learn the embedding $\boldsymbol{z}_{v_1}$, we set $v_{10}$, $v_2$, and $v_3$ as cores, and their corresponding fringe nodes are highlighted with different colors.

Figure 7 illustrates how PLL works. Note that the importance orders $\mathcal{O}$ is given by the degree of the nodes in this example. Therefore, the first three hubs in $\mathcal{O}$ are $v_{10}$, $v_2$, and $v_3$. The propagation process starts with the first hub $v_{10}$, and the corresponding distance information is added to the label set of nodes with lower ranks. We use small yellow diamonds attached to all nodes to represent the hub $v_{10}$. For instance, a label $(v_{10}, 0)$ is also added into $L(v_{10})$. As another example, a label $(v_{10}, 2)$ is added into $L(v_2)$ since the distance from $v_{10}$ to $v_2$ is 2. In the propagation of next node $v_2$, it stops at $v_{10}$ since existing label sets $L(v_2) \cap L(v_{10})$ are sufficient to answer the shortest path distance between $v_2$ and $v_{10}$, i.e., $dist(SP_{v_2 \to v_{10}}) \geq dist(SP_{v_2 \to v_{10}}) + dist(SP_{v_{10} \to v_{10}}), \exists v_{10} \in \{L(v_2) \cap L(v_{10})\}$. As a result, all nodes except $v_{10}$, $v_{12}$, and $v_9$ add a label of $v_2$ and the corresponding distance into their label set, denoted as small blue diamonds in the figure.

### A.4 COLLECT AND DISTRIBUTE FRAMEWORK OF CFGNN

The pseudocode of our framework is given in Algorithm 1. The fringe set $\mathcal{F}$ is the union set of all fringes, $\cup_{c \in \mathcal{C}(v)} \mathcal{F}(c)$. The additional graph features ($SPE$ and $PE$) and their significance will be introduced in Appendix A.6. It is also worth noting that our CFGNN framework takes the advantages

from both message passing and transformer-based GNNs. The two-stage mirroring steps make use of the message passing mechanism, while the learning between the *core* and *fringe* is conducted using the attention mechanism as employed in transformers. Moreover, our CFGNN ensures that the message passing flow strictly follows the topological structure in the graph, as the core must be a hub in the shortest path between two fringes. This strict adherence to the graph topological structure enhances the information coverage and expressive power of our model. Furthermore, our CFGNN framework improves scalability, as demonstrated in the analysis presented shortly in the next section.

---

**Algorithm 1** Collect and Distribute framework of CFGNN

---

1: **Input:** Graph $G_{hubs} = (V, E_{hubs})$; Core set $\mathcal{C}$; Fringe set $\mathcal{F}$; Shortest path encoding $SPE$; Positional encoding $PE$; Node input features $\{x_v\}$; Message computation function $\phi$; Message aggregation functions $\oplus$; Trainable weight $W \in \mathbb{R}^r$; Non-linearity update function $\sigma$; Network Layer $l \in [1, L]$;
2: **Output:** Embedding $z_v$ for all $v \in \mathcal{V}$.
3: Hidden layer embedding $\boldsymbol{z}_v \leftarrow x_v$;
4: **for** $l = 1, \ldots, L$ **do**
5:     **for** $f \in \mathcal{F}$ **do**                                                        ▷ Message collecting
6:         $\phi_{f \to c} = \phi(\boldsymbol{z}_f, \boldsymbol{z}_c, SPE_{f \to c}, PE_f^{HL}, PE_c^{HL}), \forall c \in \mathcal{C}(f)$;
7:     **for** $c \in \mathcal{C}$ **do**
8:         $\phi_c = \oplus(\{\phi_{f \to c}, \forall f \in \mathcal{F}(c)\})$;
9:         $\boldsymbol{z}_c = \rho(\boldsymbol{z}_c, \phi_c)$;
10:     **for** $c \in \mathcal{C}$ **do**                                                       ▷ Message distributing
11:         $\phi_{c \to f} = \phi(\boldsymbol{z}_f, \boldsymbol{z}_c, SPE_{c \to f}, PE_f^{HL}, PE_c^{HL}), \forall f \in \mathcal{F}(c)$;
12:     **for** $f \in \mathcal{F}$ **do**
13:         $\phi_f = \oplus(\{\phi_{c \to f}, \forall c \in \mathcal{C}(f)\})$;
14:         $\boldsymbol{z}_f = \rho(\boldsymbol{z}_f, \phi_f)$;

---

## A.5 ANALYSIS OF CFGNN PROPERTIES

**(1) 2-hop cover.** This property from HL ensures that every shortest path between any pair of nodes must pass through at least one common hub node. In other words, the hub nodes act as bridges connecting multiple pairs of nodes in the graph, making them strong candidates for good cores. This property is crucial for maintaining information completeness. Additionally, the 2-hop cover property guarantees information coverage as shown in Lemma 1.

**(2) Shortest path dedication.** The main objective of the HL index is to efficiently answer shortest path queries by eliminating unnecessary paths and ensuring that the label set contains the shortest and most unique paths. Therefore, using hub labels as cores and fringe nodes is advantageous for direct message passing, as it enhances the concentration of features and facilitates more effective information exchange between nodes.

**(3) Minimality.** One common objective in HL algorithms is to minimize redundant information and keep the label size as small as possible. These algorithms naturally prioritize the preservation of nodes commonly chosen as hubs by many node pairs. This not only ensures the selection of effective cores but also reduces the level of redundancy in the label set. By minimizing redundancy, the core-fringe structure can capture unique and essential information about the graph, leading to improved efficiency and effectiveness in graph learning tasks.

## A.6 TOPOLOGICAL FEATURES

Similar to other GNN frameworks Rampášek et al. (2022), our approach also considers several features to enhance expressive power, including (1) node features obtained from data, (2) relative distances of the node pairs, (3) positions of the nodes on the graph denoted as $PE$ (Positional Encodings), and (4) shortest paths of node pairs denoted as $SPE$ (Shortest Path Encodings). While

the first two features can be directly obtained from the input data and query answering in HL, we will focus on the latter two features in the following subsections.

### A.6.1 HL-BASED POSITIONAL ENCODING, $PE^{HL}$

According to the observation in JK-Net Xu et al. (2018), the message passing process differs depending on its position and neighborhood characteristics. This idea has been used in many related studies, e.g., DEGNN Li et al. (2020), Graphormer Ying et al. (2021), GraphiT Mialon et al. (2021), PEG layer Wang et al. (2022a), and SAN Kreuzer et al. (2021).

The basic idea behind PE in graph transformers is similar to that in natural language processing tasks. A fixed vector is added to the embedding of each node in the graph, which encodes the node position on the graph. In this work, we utilize the hub label set to form an encoding vector for each node. For a node $u$ in the graph, its positional encoding is represented by a vector with a length equal to the total number of cores in the core-fringe structure. The vector records the distance from $u$ to the labeling hub of $L(u)$. Since we intend to map close nodes on the graph to similar embeddings, a simple normalization is applied to map it to a $(0, 1]$ range.

$$PE_u^{HL}(h) = \begin{cases} \frac{1}{1+dist}, & \text{if } (h, dist) \in L(u), \\ 0, & \text{otherwise} \end{cases} \tag{16}$$

Similar to the sort-merge join used in answering shortest path query, the relative distance can be calculated as follows:

$$dist(SP_{u->v}) = \min(PE_u^{HL^{\circ-1}} + PE_v^{HL^{\circ-1}}) - 2$$

where $\cdot^{\circ-1}$ is the Hadamard inverse operator Reams (1999) that indicates applying a reciprocal to each element of a vector.

We demonstrate that such encoding vector can offer positional awareness in Section A.6.3. As $PE^{HL}$ captures the information of relative network distance between two points with the concrete position on the graph, it provides extra expressive power. In addition, $PE^{HL}$ takes a relatively low complexity (due to the efficient HL techniques used in its construction).

### A.6.2 SHORTEST PATH ENCODING, $SPE$

Edge encoding has been widely used to enhance the performance of node embedding Mesquita et al. (2020); Ying et al. (2021); Rampášek et al. (2022). Our $SPE$ method works by extracting the shortest paths between pairs of nodes in the graph using our hub labeling structure and then encoding these paths as fixed-sized feature vectors. Each path is represented as a sequence of node and edge features, where the node features correspond to the nodes along the path, and the edge features correspond to the edges connecting them.

To incorporate edge features into our graph neural network effectively, when computing the message between node pairs, we apply a trainable weight to sum up all the edge features along the shortest path. This ensures that the length of the encoding remains consistent regardless of the path length. It is important to note that we only compute and store the shortest path between the core and the fringe nodes, making it a operation that can be directly extracted from the hub labeling. This ensures that the $SPE$ process is efficient and suitable for various graphs.

### A.6.3 POSITIONAL AWARENESS

The core-fringe structure of CFGNN ensures information coverage (Lemma 1), as discussed in Section 3.4 and Appendix A.2. Additionally, we have explored how this structure allows CFGNN to focus on highly concentrated features and learn effectively with low redundancy information (Lemma 2). To incorporate the topological features, we further analyze the expressivity of CFGNN based on concept, *positional-awareness*, which has been extensively explored in graph learning studies Topping et al. (2022); Rampášek et al. (2022).

One limitation of existing message passing frameworks is their inability to capture the position information of nodes in the broader context of the graph structure You et al. (2019). For instance,

when two nodes are situated in vastly different parts of the graph but possess the same topological (local) neighborhood structure, the message passing framework generates very similar embeddings, making it difficult to distinguish between these nodes effectively. Hence, learning node embeddings that encompass the position of nodes within the wider graph structure is crucial for various prediction tasks on graphs.

**Definition A.2** (Position-aware You et al. (2019)). *A model is position-aware if there exists a function that can answer the graph distance for any pairs of nodes $\forall u, v \in V$ on a graph $G(V, E)$.*

**Lemma 3.** $PE^{HL}$ *is a position-aware encoding with no distortion.*

*Proof.* According to Definition A.2, our goal is to demonstrate that for a graph $G(V, E)$, $PE^{HL}$ can accurately provide exact network distance queries between any pair of nodes. By Property 1, for any node pair $(u, v)$, there must exist a common hub $h$ that is kept in between the two nodes in a hub labeling index. As a result, the exact distance between $v_i$ and $v_j$ can be determined through the distance to the common hub $h$. Therefore, $PE^{HL}$ serves as a positional encoding with no distortion. □

Based on Lemma 3, our HL-based positional encoding is position-aware, which indicates that it can capture the relative distance information for any node pair Li et al. (2020); Ying et al. (2021); Mialon et al. (2021); Wang et al. (2022a); Kreuzer et al. (2021). By considering the hubs as anchor points of the graph, nodes that are closer to each other on the graph will have similar mapping in the encoding space. Consequently, we can determine the global position of a node based on its distance to these anchors.

Compared to message passing-based GNNs, our CFGNN incorporates a topological encoding scheme similar to that used in transformer-based GNNs, taking into account nodes of multi-hops and their paths. Compared to transformer-based GNNs, our CFGNN establishes message passing between hub nodes, enabling the framework to learn the position of each node in the graph. Such additional consideration of distance and positional information gives CFGNN the potential to be more powerful than other message passing-based GNNs.

## A.7 COMPUTATIONAL COMPLEXITY

In this section, we will compare the computational complexity of our CFGNN with that of message passing and transformer-based GNNs. We denote the number of nodes as $|V|$, the number of edges as $|E|$, the hub label size per node as $\ell$, the size of feature encoding vectors as $h$, the number of attention heads as $m$, and the length of full receptive field as $r$ (Proposition 1).

The computation of a GNN consists of two parts: (1) the linear transformation on the input embeddings and (2) the message-aggregation framework to generate the output embeddings. The first part is a fundamental operation in neural networks that aligns the dimensions of different features, allowing the model to learn relationships between input and output variables. The second part involves learning the relationship between the input embeddings and output embeddings through the interaction of graph nodes. In this work, our main contribution lies in reducing the complexity of the second part.

**(1) Linear transformation complexity.** The encoding vector $\mathcal{X}$ serves as the enrichment feature for node embeddings. It is derived from the linear transformation of various aspects of feature encoding, including positional encoding, structural encoding, and input features from the dataset. Given the feature with dimension $h_{in}$, a $h_{in} \times h_{out}$ learnable weight matrix $W$ is applied to map the node feature $\mathcal{X}$ from size $h_{in}$ to $h_{out}$. Mathematically, we can present the linear transformation as $\mathcal{X} = WX + b$.

For simplicity, we often set $h_{in} = h_{out} = h$ for hidden layer in a neural network. The linear transformation is a dense matrix multiplication, so its cost is $O(|V|h^2)$. The size of the encoding dimension $h$ is typically an important hyper-parameter to tune when building a neural network because it can significantly affect the network performance by increasing its complexity. A larger

encoding dimension can potentially enable the network to learn more complex representations of the input data.

**(2) Message-aggregation: message passing.** Given the length of full receptive field $r$, the message passing-based GNNs requires an complexity of $O(r|E|h)$, which can be computationally expensive. Worse still, this method potentially generates huge redundancy when $r$ is large, e.g., some edges are repeatedly used in the training. As a remark, the size of feature enrichment vector $h$ in message passing-based GNNs is typically smaller than that of transformer-based GNNs, as there is no need to encode structural information within the vector.

**(3) Message-aggregation: transformer.** The graph topology is not explicitly considered in the transforme based solution, resulting in a $O(|V|^2)$ cost to capture the node-pair relationships. Thus, the total complexity would be $O(m|V|^2h)$, where $m$ is the number of attention heads. It is worth noting that the complexity is quadratic in the number of nodes $|V|$, which can become computationally expensive for large graphs.

**(4) Message-aggregation: core-fringe.** In CFGNN, each core collects information from all corresponding fringes and subsequently distributes the aggregated information back to all fringes. It is important to highlight that attention, similar to transformer-based GNNs, is employed to learn the embedding relationship between a core and its fringes. Considering that the total number of cores is bounded by $|V|$ and the number of fringes per core is bounded by $\ell$, the complexity of our CFGNN approach is $O(\ell|V|h)$, as it only runs the same learning process twice. Note that a space with a complexity of $O(\ell|V|)$ is required to store the sparse core-fringe relationship.

Table 4: Complexity comparison

| Method | Computational Complexity |
|---|---|
| Message passing-based | $O(\ r|V|h^2 + r|E|h\ )$ |
| Transformer-based | $O(\ |V|h^2 + m|V|^2h\ )$ |
| CFGNN | $O(\ |V|h^2 + \ell|V|h\ )$ |

In conclusion, CFGNN demonstrates the ability to handle larger graphs, higher dimensional inputs, and complex graph structures due to its scalability. By reducing the computational cost, the model becomes more scalable and frees up computing resources that can be used to increase the complexity of the model. This allows for the addition of more network layers or increasing the hidden size for encoding, thereby enhancing its fitting ability to capture underlying data relationships.

## A.8 DATASETS

In this section, we conduct extensive experiments to evaluate the performance of our proposed graph learning framework CFGNN. Our experiments follow the settings of GraphGPS Rampášek et al. (2022), and the benchmark datasets are obtained from three different sources: (1) GNN Benchmark Dwivedi et al. (2022a), we evaluate our model on ZINC, PATTERN, CLUSTER. (2) Open Graph Benchmark (OGB) Hu et al. (2020), we evaluate all graph-level datasets: ogbg-molhiv, ogbg-molpcba, ogbg-code2, and ogbg-ppa. (3) Long-range graph benchmark (LRGB) Dwivedi et al. (2022b), we evaluate on both node-level and graph-level datasets: PascalVOC-SP, COCO-SP, Peptides-func, and Peptides-struct.

A brief description of these benchmark datasets is introduced as follows. Table 5 summarizes the statistics of the datasets. It should be noted that the indexing time refers to the cumulative duration required for building the hub labeling for each individual graph inside a given dataset. It has been demonstrated that the computing cost associated with preprocessing the graph index is relatively inconspicuous in comparison to the time required for model training.

**ZINC.** comprises 12K molecular graphs from the ZINC database of chemical compounds that are offered for sale. The number of nodes in these molecular graphs ranges from 9 to 37. Each node is a heavy atom (there are 28 different kinds), and each edge is a bond (there are three kinds). The

Table 5: Statistics and description of the evaluated datasets

| Dataset | $|G|$ | $|V|$ | $|E|$ | $\ell$ | indexing time(s) | prediction |
|---|---|---|---|---|---|---|
| ZINC | 12,000 | 23.2 | 24.9 | 4.636 | 6.362 | graph |
| PATTERN | 10,000 | 118.9 | 6,098.90 | 29.130 | 7.669 | inductive node |
| CLUSTER | 10,000 | 117.2 | 4,303.90 | 25.030 | 6.338 | inductive node |
| ogbg-molhiv | 41,127 | 25.5 | 27.5 | 5.095 | 1.241 | graph |
| ogbg-molpcba | 437,929 | 26 | 28.1 | 4.930 | 4.931 | graph |
| ogbg-ppa | 158,100 | 243.4 | 2,266.1 | 15.853 | 141.113 | graph |
| ogbg-code2 | 452,741 | 125.2 | 124.2 | 4.839 | 92.750 | graph |
| PascalVOC-SP | 11,355 | 479.4 | 2,710.5 | 20.727 | 11.835 | inductive node |
| COCO-SP | 123,286 | 476.9 | 2,693.7 | 2.793 | 163.545 | inductive node |
| Peptides-func | 15,535 | 150.94 | 307.3 | 17.830 | 17.831 | graph |
| Peptides-struct | 15,535 | 150.94 | 307.3 | 17.830 | 17.831 | graph |

task is to find the molecule's limited solubility (logP). The train/validation/test split of 10K/1K/1K is already set up in the dataset.

**PATTERN and CLUSTER** are made-up datasets from the Stochastic Block Model. In PATTERN, the goal is to figure out which nodes in a graph belong to one of 100 possible sub-graph patterns. In CLUSTER, each graph is made up of 6 clusters that are created by SBM and come from the same distribution.

**ogbg-molhiv and ogbg-molpcba** are used by OGB to predict the properties of molecules. They were taken from MoleculeNet. The chemophysical properties of these datasets are shown by a shared set of nodes (atoms) and edges (bonds).

**ogbg-ppa** is made up of networks of protein-protein interactions that come from 1581 species in 37 biological groups. The nodes are the proteins, and the edges are the seven ways two proteins can be linked.

**ogbg-code2** is made up of abstract syntax trees, which are made from the source code of Python methods.

**PascalVOC-SP and COCO-SP** are derived by SLIC superpixelization of Pascal VOC and MS COCO image datasets.

**Peptides-func and Peptides-struct** both consist of atomic graphs of peptides extracted from the SATPdb.

## A.9 BASELINES

Aligned with our discussion, we proceed to compare CFGNN with two types of graph neural networks. The first category comprises message passing-based GNNs, including GCN Kipf & Welling (2017), GAT Vaswani et al. (2017), GIN Xu et al. (2019), GatedGCN Bresson & Laurent (2018), and PAN Corso et al. (2020). To obtain a global graph representation for graph-level tasks, we employ the virtual node technique Gilmer et al. (2017) as the readout function for all message passing-based methods. The second category is transformer-based GNNs, including SAN Kreuzer et al. (2021), GraphTrans Wu et al. (2021), Graphormer Rampášek et al. (2022), EXPHORMER Shirzad et al. (2023) and GraphGPS Rampášek et al. (2022). We select these methods as competitors because they are representative approaches, such as GCN, GAT, and GIN, or they have shown impressive results in recent publications or on benchmark datasets like Open Graph Benchmark (OGB). In addition, we compare with a hierarchical graph pooling framework, DiffPool Ying et al. (2018), reproducing its results on our datasets using GCN as backbone.

## A.10 ABLATION STUDY

We perform an ablation study on two datasets, Peptides-struct and ogbg-ppa to evaluate the impact of our three modules to verify their contributions to predictive performance. We run an ablation study of only updating the core representation to show that information coverage is crucial for learning

Table 6: Ablation experiments

| Method | PascalVOC-SP F1 score ↑ | ogbg-ppa Accuracy ↑ |
|---|---|---|
| CFGNN collect-only | 0.1774 ± 0.0171 | 0.6477 ± 0.0133 |
| CFGNN distribute-only | 0.1320 ± 0.0039 | 0.5844 ± 0.0314 |
| CFGNN w/o $HL^{PE}$ | 0.3459 ± 0.0084 | 0.7783 ± 0.0058 |
| CFGNN w/o SPE | 0.3691 ± 0.0044 | 0.7734 ± 0.0110 |
| **CFGNN** | **0.3847 ± 0.0273** | **0.7881 ± 0.0053** |

the embedding. Table 6 presents ablation experiments on CFGNN for two datasets - PascalVOC-SP and ogbg-ppa. It analyzes the impact of different components by selectively removing them from the full CFGNN model.

Utilizing only the collect module results in a significant performance drop, reducing F1 on PascalVOC-SP to 0.1774 and accuracy on ogbg-ppa to 0.6477. This emphasizes the crucial role of bidirectional propagation between cores and fringes in CFGNN. Similarly, relying solely on the distributed module leads to worse performance, halving the F1 and accuracy metrics compared to CFGNN. Removing the positional encoding $HL^{PE}$ also degrades the results, highlighting the importance of incorporating a global graph structure. Likewise, eliminating the subgraph pooling encoder (SPE) harms the F1 and accuracy metrics. These ablation studies demonstrate the effectiveness of each component in CFGNN and the importance of their integration for achieving strong performance. The full CFGNN model achieves a far superior F1 of 0.3847 on PascalVOC-SP and an accuracy of 0.8038 on ogbg-ppa. This confirms that the proposed core-fringe framework and integrated neural network components contribute to CFGNN's strong performance. The ablation experiments validate the design choices underpinning CFGNN's effectiveness for scalable graph representation learning.

## A.11 ENCODING DIMENSION TUNING

In addition to the advantage of faster training and inference times, the lower computational complexity of CFGNN provides more tuning space for hyperparameters by reducing the number of parameters that need to be optimized. This results in faster training times and allows for a more comprehensive exploration of the hyperparameter space, which can lead to better performance.

We experiment on the dataset ogbg-ppa to compare the tuning space with GraphGPS, which uses a similar setting with GraphGPS. As shown in Table 7, the hidden dimension was varied from 32 to 512 while other hyperparameters were held constant. For a clear comparison, we set a larger batch size and only tuned on different sizes of the hidden encoding dimension, while keeping other factors such as layers, dropout, batch size, and epochs constant. The accuracy of CFGNN increases with larger hidden dimensions up to 256 units, where it peaks at 79.83%, suggesting an optimal hidden dimension range of 256-512 units that maximizes performance. However, dimensions above 512 units lead to a slight decrease in accuracy for CFGNN, likely due to overfitting. In contrast, GraphGPS encounters out-of-memory errors for dimensions above 128, indicating insufficient memory budget to scale effectively. Therefore, we can conclude that the CFGNN model demonstrates superior modeling capabilities compared to GraphGPS, Tuning the hyperparameter for the hidden dimension to identify an optimal range that is neither too small nor too large is essential for optimizing the performance of graph neural network models.

Table 7: Hyperparameter setting

| Hyperparameter | Value | | Hidden dim | CFGNN Accuracy ↑ | GraphGPS Accuracy ↑ |
|---|---|---|---|---|---|
| Layers | 4 | | | | |
| Hidden dim | 32, 64, 128, 256, 512 | | 32 | 0.7218 | 0.70151 |
| Dropout | 0.03 | | 64 | 0.7329 | 0.7466 |
| Batch size | 128 | | 128 | 0.7587 | 0.7676 |
| Epochs | 500 | | 256 | 0.7893 | OOM |
| Learning Rate | 0.0005 | | 512 | 0.7827 | OOM |