# OpenReview forum: "Bridging Indexing Structure and Graph Learning: Expressive and Scalable Graph Neural Network via Core-Fringe"
_ICLR.cc/2024/Conference — ICLR 2024 Conference Withdrawn Submission_

### Official Review · Reviewer_PUcW · 2023-10-25

**Soundness:** 3 good
**Presentation:** 4 excellent
**Contribution:** 2 fair
**Rating:** 6
**Confidence:** 4

**Summary:**

The authors question whether a GNN model with both scalability and expressivity in learning graph information exists, and propose to use a new message-passing paradigm based on hub labeling. The proposed paradigm only maintains the message-passing between core nodes and their corresponding fringe, which reduces the amount of message-passing operation in regular GNN, while cover all information in the graph. Experimental results show that the method is comparable to existing approach.

**Strengths:**

The motivation and intuition behind the hud labeling based message-passing is well explained. Overall, the paper is easy to follow with informative illustrations highlighting the proposed methods against existing approaches.

This approach significantly reduces message-passing operations required for distant nodes to read information from each other.

The approach seems well-attuned to large graphs, and experimental results show that the approach is promising compared to existing ones that potentially require more computations.

**Weaknesses:**

- The message passing is only between the core and fringe, meaning that none of the local structural information is kept. However, these structures are very important for some applications (including molecular graphs which are the main targets of the work).

- While the performance is on par with the existing approaches, CFGNN is not the clear winner. This is fine, as I believe the key contribution is that CFGNN requires much less computation resources. However, no experimental results verify the claim. (Except transformer model OOM on some datasets that CFGNN can process)

- The hub labeling process is not coupled with the learning process, meaning that the core/fringe assignment can be irrelevant to the actual learning task, which potentially cause the model to ignore information.

- While using the hub core nodes to aggregate information seems natural, it could be detrimental. For example, supernodes exist in the graph. Then, at the end of the message passing, all nodes will have very similar embedding as their messages are all from supernodes. In such cases, node embeddings will be substantially over-smoothed. In contrast, a regular GNN, because every node keeps local connectivity and has message from non-supernodes, is less affected.

**Questions:**

- How do the actual core nodes look like? Are they supernodes? If they possess similar properties across different datasets, then is it valid to use the same tool to choose core nodes for different data?

- Why do transformer models cause OOM? If you shrink the batch size, you should still be able to train the model.

Also, weaknesses mentioned above.

---

> ### Author Response · Authors · 2023-11-22
> **Response to Reviewer PUcW (1/2)**
>
> ### W1: The message passing is only between the core and fringe, meaning that none of the local structural information is kept. However, these structures are very important for some applications (including molecular graphs which are the main targets of the work).
>
> The deficiency of transformer-based models is in their inability to incorporate local structural information. Furthermore, **applying message passing among all node pairs fails to capture the structural information as well**.  As we summarized in section 2.2: “**Nevertheless, the self-attention module overlooks the crucial topological structure between nodes. In other words, the relationship between node pairs is solely captured based on their individual node features**.” Previous graph transformer methods enhanced the structural features of nodes, by **adding different encodings to the nodes themselves**. such as positional information, neighbor counts, and triangle enumeration. The local structural information you mentioned is the result of traditional message passing framework. Typical message passing methods for learning structural knowledge contain a significant amount of redundant message flow [1]. Our proposed shortest path encoding method ensures that the model can learn the structual information from all edges on the shortest path for each node pair, as such, message redundant can be reduced. In addition, as a GNN framework, any encoding that enhances node or edge features can be directly incorporated into our model. **For example, performing graph convolution on the low-level neighbors of each node as a encoding** allows us to explicitly express the local information which you mentioned.
>
> ### W2: While the performance is on par with the existing approaches, CFGNN is not the clear winner. This is fine, as I believe the key contribution is that CFGNN requires much less computation resources. However, no experimental results verify the claim. (Except transformer model OOM on some datasets that CFGNN can process)
>
> Our proposed work aimed to demonstrate that **high-quality results in graph tasks can be achieved without the need for extensive computational resources**. In addition, we conducted a comparative experiment with GraphGPS to demonstrate the reduction in computational cost reflected in fine-tuning the hidden embedding dimensions. The GraphGPS has memory exhaustion while using a hidden size of 256, however, our approach enables us to expand it to 512.
>
> ### W3: The hub labeling process is not coupled with the learning process, meaning that the core/fringe assignment can be irrelevant to the actual learning task, which potentially cause the model to ignore information.
>
> The objective of our approach is to utilize a conventional graph analysis technique, namely graph indexing, to direct the training of graph neural networks.   Given the lack of interpretability and the substantial computational resources needed, **we are reluctant to rely only on fitting data to neural networks to train all the learning modules**. The introduction of hub labeling is intended to leverage its two-hop complete coverage property to enable our model to efficiently acquire useful information instead of directly using self-attention for learning. This is also the objective of our work, "**Bridging Indexing Structure and Graph Learning**", and we aim to motivate future researchers to investigate the benefits of conventional graph analysis techniques and employ them in contemporary graph neural networks.

---

> ### Author Response · Authors · 2023-11-22
> **Response to Reviewer PUcW (2/2)**
>
> ### W4: While using the hub core nodes to aggregate information seems natural, it could be detrimental. For example, supernodes exist in the graph. Then, at the end of the message passing, all nodes will have very similar embedding as their messages are all from supernodes. In such cases, node embeddings will be substantially over-smoothed. In contrast, a regular GNN, because every node keeps local connectivity and has message from non-supernodes, is less affected.
>
> Over-smoothing is a problem in graph neural network (GNN) learning. Traditional graph convolution methods continually aggregate neighborhood information, causing the features of all nodes in a connected graph to become increasingly similar as the depth of the GNN increases. In our method, we limit **the number of message passing layers to only two**. Our approach propagates messages along the shortest paths, which is more efficient compared to the traditional methods that propagate redundant messages throughout the graph.
>
> Furthermore, although different nodes' core sets may share some core nodes, typically, the core sets of nodes at different positions in the graph are not completely identical. This means that not all nodes receive information from the same set of core nodes (as demonstrated in Appendix 3). **This brings uniqueness to the features of the nodes**, further alleviating the over-smoothing problem. Additionally, we introduce positional encoding (as described in Appendix 6.3) and other information that introduces unique identifiers for each node. With the propagation of information from the core nodes, the positional encoding provides relative distance information. Despite the existence of nodes that share core nodes, their received information differs based on their positions.
>
> Overall, our method makes several efforts to mitigate the impact of over-smoothing from multiple perspectives. The primary challenge we face is over-squashing (as described in Appendix 2). Due to the nature of the hub labeling construction method we employ, the top-ranked node is labeled by all nodes, which means it receives and propagate information from all the nodes. For that node, the information it collects may be excessively compressed, leading to a decrease in its expressive power. **We theoretically prove that the over-squashing phenomenon in our method is lower than that of message-passing-based GNN methods.**
>
> ### Q1: How do the actual core nodes look like? Are they supernodes? If they possess similar properties across different datasets, then is it valid to use the same tool to choose core nodes for different data?
>
> In our implementation, we observed that the core sets with a significant number of fringes predominantly consist of nodes with a high degree. This is because we primarily utilize the degree of nodes as the criterion for constructing hub labeling. As stated in **Appendix 5(original submission)**, our assessment of the validity of the core-fringe construction is based on three properties: **(1) 2-hop cover, (2) Shortest path dedication, and (3) Minimality**. The heuristic hub pushing construction method satisfies these requirements for most realistic complex networks, indicating its adaptability to different types of datasets.
>
> ### Q2: Why do transformer models cause OOM? If you shrink the batch size, you should still be able to train the model.
>
> Currently, in the graph task benchmarks [2,3], the size of the graph consists of several hundred nodes. By reducing the batch size and hidden size, it is possible to make previously difficult training methods fit in GPU memory. However, this comes at the cost of reducing the efficiency and capability of the model training. For example, reducing the batch size significantly increases the training time, while reducing the hidden size may lead to the problem of over-squashing (refer to Appendix 2). The lower the difficulty of model training, the larger the space for fine-tuning the model, as shown in Appendix 11. That is why we preserve the original settings when comparing our method with baseline methods.
>
> [1] Redundancy-Free Message Passing for Graph Neural Networks. Nips 2022
>
> [2] Open Graph Benchmark: Datasets for Machine Learning on Graphs. NeurIPS 2020
>
> [3] Long Range Graph Benchmark. NeurIPS 2020

---

> ### Comment · Reviewer_PUcW · 2023-11-22
>
> After reading the author's response, some of my concerns are addressed, including the claim about computational efficiency and analysis on over-smoothing. However, I still believe more quantitative analysis should be provided to validate the claimed efficiency. Some concerns are not fully addressed. For example, the proposed method still does not capture local topology well. I believe the reference [1] is not about information redundancy but computational redundancy; it keeps the same information as a message-passing GNN, while the proposed method explicitly drops local edge information. Also, the problem of using one hub-labeling method for all data remains, and combining the "conventional graph analysis technique" with GNN idea is not entirely novel.
>
> Overall, the paper presents an interesting and well-illustrated method to reduce the computational cost. Yet, the method heavily relies on the hub-labeling trick, which is not data-dependent and, in turn, limits the contribution. Hence, I am keeping my initial rating.

---

### Official Review · Reviewer_T5WA · 2023-10-31

**Soundness:** 2 fair
**Presentation:** 3 good
**Contribution:** 1 poor
**Rating:** 3
**Confidence:** 3

**Summary:**

The authors proposes a new way of aggregating messages in graph neural networks, called CFGNN.

CFGNN divides nodes into core and fringe groups based on hub labeling.

CFGNN first propogates node embeddings from the fringe groups into the core nodes, then propogates node embeddings from the code nodes to the corresponding fringe groups.

This technique scales the receptive fields of the GNN such that the receptive field is not linear to the number of layers. CFGNN is more computationally efficient than transformers, as it applies sparse message passing along core and fringe nodes. The authors evaluate their approach against 7 datasets and achieve superior performance.

**Strengths:**

How to extend GNNs receptive field and scalability are both important questions in the field.

The authors propose a model which is scalable and theoretically has large receptive field and beats baselines.

The authors carefully detail their motivations throughout the paper.

**Weaknesses:**

I have several issues with the work:

1) Novelty
- The paper essentially takes the Graph U-Net[1] approach of pooling and unpooling node embeddings from a core subset of node embeddings, yet the authors do not mention this work.
- Unlike Graph U-Net[1], the core subset of node embedding are determined by hub labeling rather than being learned. I believe this would hurt CFGNN's performance given sufficient data to learn the hub labeling.
- There have been several related works addressing receptive fields of GNNs, which are not compared against: Graph U-Net[1], Deeper GNNs[2], Multihop GNNs[3].
- As a suggestion, I implore the authors to look more into how indexing can improve the scalability of their approach over said existing techniques. However, I believe more explanation and experiments are needed to distinguish this work from others in this aspect.

2) Experiments
- There are GNN baselines [1,2,3] that address the receptive fields issue, which the authors do not compare against.
- The authors do not compare against scalable deep GNN techniques such as sampling[4,5] or distillation[6].
- Given CFGNN tradeoffs the expressive power of graph transformers for increased scalability, the paper lacks scalability studies between CFGNN and Graphnormer, which would clarify when one should be used over the other.
- I am unsure what the * mean?
- There is a lack of ablation study (ex. alternatives to hub labeling for choice in core nodes, how many hub nodes?, etc.)

3) Writing
- I am unsure of the hub labeling notation: what is the difference between c and v in "L(c) and L(v)" in the section "HL-based core-fringe structure"?
- Sections 2.1 and 2.2 could be shortened. Much of the discussion in these sections are well-known (definition of receptive field, linear scalability of receptive field and layer count, and neighborhood definitions of transformers). I appreciate the author's description of these concepts, but believe this section can be greatly shortened to leave more space for the author's own proposals.

References:

[1] Graph U-Nets (ICML2019)

[2] Towards Deeper Graph Neural Networks (KDD2020)

[3] Multi-hop Attention Graph Neural Network (IJCAI2021)

[4] Inductive Representation Learning on Large Graphs (NeurIPS2017)

[5] Training Graph Neural Networks with 1000 Layers (ICML2021)

[6] Graph-less Neural Networks: Teaching Old MLPs New Tricks Via Distillation (ICLR2022)

**Questions:**

Please see the weaknesses section. I am mainly concerned with:
1) What is the relationship between CFGNN and Graph U-Net? Are there any empiracal comparisons between the two?
2) What is the relationship between CFGNN and scalability techniques applied on regular GNNs/transformers?
3) What is the scalability relationship between CFGNN and transformers? Are there any empiracal comparisons?

---

> ### Author Response · Authors · 2023-11-22
> **Response to Reviewer T5WA (1/3)**
>
> ## W1 (Novelty):
>
> ### *a) The paper essentially takes the Graph U-Net approach of pooling and unpooling node embeddings from a core subset of node embeddings, yet the authors do not mention this work.*
>
> Our approach differs from the method using pooling and unpooling operations in Graph U-Net. The pooling structure is a **multi-layer graph information compression process** aimed at increasing the receptive field of each node. In contrast, our method, as stated in Section 2.1, is a **novel graph propagation framework that is theoretically proven to achieve global attention with only two layers**. Furthermore, we have also expanded the definition of receptive field, allowing our framework to simultaneously consider path features among nodes during the information propagation process.
>
> Besides, our method emphasizes the **comprehensive consideration of node features and path features, which were overlooked by previous graph pooling methods such as U-Nets and DiffPool**. Our goal is to ensure that the model not only considers node features but also emphasizes the features on the corresponding edges of the paths. This feature is not commonly found in traditional pooling and unpooling methods.
>
> Additionally, our method introduces **the concept of core nodes**, which play a transition role in information propagation. Through these **core nodes**, our method guarantees **complete information coverage for all nodes in just two steps**. This means our method can receive information from all points on the graph and includes relevant information from the shortest path. In contrast, hierarchical pooling methods do not provide such theoretical guarantees and require more layers to achieve similar effects, and the number of layers is uncertain.
>
> ### *b) Unlike Graph U-Net[1], the core subset of node embedding are determined by hub labeling rather than being learned. I believe this would hurt CFGNN's performance given sufficient data to learn the hub labeling.*
>
> As stated in Section 3.2, We aim to provide more interpretability and efficiency while maintaining model performance. We recognize the advantages of traditional graph analysis techniques in interpretability and their computational resource requirements. **We firmly believe that combining traditional graph analysis techniques with modern graph neural networks can yield better results and bring new perspectives and directions to the field of graph learning.** In this regard, our proposed method, CFGNN, has unique advantages. This framework achieves complete information coverage with a two steps framework and introduces a core-fringe structure that requires carefully designed algorithms to ensure the **theoretical guarantee** of **2-hop coverage**. This design cannot be directly obtained through machine learning, which is precisely why our method stands out from ordinary hierarchical models.
>
> We believe that the proposed framework, which **combines data structure techniques and modern machine learning approaches**, can achieve better results in graph analysis. It maintains model performance and provides more interpretability and computational efficiency. We hope the proposed approach can bring new perspectives and directions to the field of graph analysis and provide more effective solutions for solving real-world problems.
>
> ### c) There have been several related works addressing receptive fields of GNNs, which are not compared against: Graph U-Net, Deeper GNNs, Multihop GNNs
>
> In Section 2, we introduce "**information coverage**" as a new concept, distinct from the previously mentioned receptive field. We ensure the model extracts relevant information from at least one shortest path, maximizing information coverage. The core value of this concept lies in effectively capturing important information in the graph structure when attempting to have global attention. At the same time, we recognize that the three mentioned methods fail to address this issue as they overlook the information contained in paths within the graph structure. The three methods can be seen as different approaches to extend the receptive field (**hyper node pooling, deepening layer, and extending adjacency matrix respectively**). While they may possess a comprehensive receptive field according to the traditional definition, none of them are capable of retaining structural information coverage such as edge, path or position.

---

> ### Author Response · Authors · 2023-11-22
> **Response to Reviewer T5WA (2/3)**
>
> ### d) As a suggestion, I implore the authors to look more into how indexing can improve the scalability of their approach over said existing techniques. However, I believe more explanation and experiments are needed to distinguish this work from others in this aspect.
>
> We appreciate the reviewers' attention to our method and are grateful for the opportunity to clarify and further explain the uniqueness and advantages of our approach.  As stated in Section 2, we employed the indexing technique known as hub labeling to construct a 2-hop cover graph. This allows the nodes to have a global attention by propagating messages over a 2-stage message passing framework. The proposed framework significantly reduce the computation cost compare to self attention transformer, making it more scalable.
>
> Our research goes beyond proposing a mere method, it opens up an entirely new research direction in graph learning. The integration of traditional graph analysis techniques with modern graph neural networks holds significant importance for advancing the field of graph analysis. We strongly believe that this fusion can bring forth a wealth of innovative ideas and research possibilities while offering novel solutions to complex graph analysis problems.
>
> ## W2 (Experiment):
>
> ### a) There are GNN baselines that address the receptive fields issue, which the authors do not compare against.
>
> Our primary criterion for selecting a baseline for experimentation is their performance on widely recognized benchmark datasets such as **OGB [2]** and **LRGB [3]**. We have noticed that all of the models mentioned by the reviewers did not submit results on these datasets.
>
> **Diffpool was chosen as the representative of a hierarchical pooling and unpooling graph learning framework in our comparison**. It exhibits poor performance on the baselines. In fact, based on our experimental results (refer to Tables 1, 2, 3, and 5), message-passing-based methods are generally dominated by transformer-based methods on our benchmark datasets. In Section 2.2, we discuss the limitations of message passing methods and highlight the need for broader information coverage, which involves considering not only absorbing more node features but also preserving more structured information, such as ensuring the existence of at least one simple path between nodes. Without such considerations, there is a risk of losing crucial topological information, leading to subpar performance on **graph tasks**. The three methods mentioned by the reviewers are all based on message passing (for **node level tasks**), which fails to capture the structural information we mentioned, such as paths and positional information on the graph.
>
> ### b) The authors do not compare against scalable deep GNN techniques such as sampling or distillation.
>
> Sampling methods are typically applied to node-level problems. In such problems, the graph can scale up to millions of nodes, and the quadratic complexity of Transformer models makes them infeasible for large-scale graphs. This can be evidenced by the lack of submissions by Transformer-based methods on the Open Graph Benchmark (OGB) node task leaderboard. In contrast, message-passing methods, using techniques like sampling and distillation, can maintain a constant complexity level, regardless of the graph's size.
>
> Our method, like other Transformer-based methods, focuses on addressing graph-level tasks. In such tasks, a dataset may contain hundreds of thousands of graphs, each with hundreds of nodes. However, any sampling method introduces information loss, such as discarding nodes or edges. Therefore, given sufficient computational resources, direct full-batch training with message passing proves superior. That is why, in our experiments, all message-passing-based methods were trained without introducing any sampling or distillation techniques.
>
> ### c) Given CFGNN tradeoffs the expressive power of graph transformers for increased scalability, the paper lacks scalability studies between CFGNN and Graphnormer, which would clarify when one should be used over the other.
>
> We would like to clarify that our work does not involve a tradeoff based on graph transformers and we do not assert that CFGNN is intended to supplant Graphormer. As outlined in Section 1, our research introduces a novel two-stage learning framework that **achieves complete information coverage in global attention**, **in a more efficient manner**. We extend the scalability of the model by reducing complexity of self attention from quadratic to sub-quadratic, while preserving the desired expressive power of the model, as analyzed in Section 2.
>
> ### d) I am unsure what the * mean?*
>
> Due to space limitations, we only provided the introduction of baseline results in Appendix 9 **(original submission)**. In the second paragraph, we used an asterisk (*) to highlight that these results were excerpted from the original papers.

---

> ### Author Response · Authors · 2023-11-22
> **Response to Reviewer T5WA (3/3)**
>
> ### e) There is a lack of ablation study (ex. alternatives to hub labeling for choice in core nodes, how many hub nodes?, etc.)
>
> The labeling technique is not just employed for the selection of hyper nodes or virtual nodes, as is commonly done in other pooling methods.It aims to find the most efficient to propagate information among all node pairs with the hubs playing the transition role. As stated in Appendix 5 **(original submission)**, the construction of the hub labeling is determined by the labeling method in order to satisfy the three properties. The number of hubs or the quantity of labeling is predetermined and cannot be altered without compromising the theoretical guarantee. This is the reason why we did not provide an ablation study on adjusting the quantity or selection of hub nodes,.
>
> ## W3(writing):
>
> ### a) I am unsure of the hub labeling notation: what is the difference between c and v in "L(c) and L(v)" in the section "HL-based core-fringe structure"?
>
> Thank you for pointing out this issue, and we apologize for the symbol error in that section. We made the necessary corrections accordingly.
>
> ### b) Sections 2.1 and 2.2 could be shortened. Much of the discussion in these sections are well-known (definition of receptive field, linear scalability of receptive field and layer count, and neighborhood definitions of transformers). I appreciate the author's description of these concepts, but believe this section can be greatly shortened to leave more space for the author's own proposals.
>
> In Section 2.1, we introduced the concept of Information coverage, which differs from the concept of prior defined receptive fields [1]. Previous definitions of receptive fields only considered the contributions of individual nodes and did not consider structural information such as paths and edges in the graph. Therefore, our proposed Information coverage provides a more comprehensive understanding of the role of models in analyzing the expressive power of graph structural information.
> Since our method involves both message passing in Graph Neural Networks and Transformer-based approaches, we dedicated a significant portion of Section 2 to provide a detailed introduction to these two methods and compare their expressive power and scalability regarding graph coverage. We did so to clearly elucidate the motivations and advantages of our method's design.
>
> We sincerely appreciate the reviewers' valuable feedback. **We consider shortening the section in the revised version while ensuring a clear explanation that allows the reviewers to appreciate the value presented in this section.**
>
> The answer to the questions from reviewer is stated in W1 and W2.
>
> [1] A Brief Review of Receptive Fields in Graph Convolutional Networks. WI 2019
>
> [2] Open Graph Benchmark: Datasets for Machine Learning on Graphs. NeurIPS 2020
>
> [3] Long Range Graph Benchmark. NeurIPS 2020

---

### Official Review · Reviewer_iyTY · 2023-11-08

**Soundness:** 2 fair
**Presentation:** 2 fair
**Contribution:** 2 fair
**Rating:** 5
**Confidence:** 5

**Summary:**

The overall research problem is interesting and the proposed method is easy to follow and understand. However,  some necessary technical details are missing. The experiment seems weak. The empirical and theoretical analysis of scalability is not adequate. Please refer to the following sections for more details.

**Strengths:**

- The research problem is interesting and pragmatic.

- The related work discussion is comprehensive.

- The experimental setting is extensive.

**Weaknesses:**

- The experiments seem weak, the experimental results are not competitive, and the selected baselines are not state-of-the-art [1].

- Motivated by the scalability, but the selected datasets are not large-scale, and the time complexity analysis is not formal.

- Section 2.1 is not informative, and its relation with the following proposed method is not clear.

- It is better to add necessary contexts for proposed equations.

- It is better to add references for the selected benchmarks and baselines.

[1] Hamed Shirzad, Ameya Velingker, Balaji Venkatachalam, Danica J. Sutherland, Ali Kemal Sinop. Exphormer: Sparse Transformers for Graphs. ICML 2023

**Questions:**

How core nodes are selected in the proposed method?

In Eq. 3, how the matrix $\mathbb{A}_{N}$ is constructed?

In Eq. 4, how the matrix $\mathbb{A}_{L}$ is constructed?

---

> ### Author Response · Authors · 2023-11-22
> **Response to Reviewer iyTY (1/2)**
>
> ## W1: The experiments seem weak, the experimental results are not competitive, and the selected baselines are not state-of-the-art [1].
>
> In our paper, we chose GraphGPS and SAN as the main baselines, as they were among the most representative methods in our research field at the time. In our comparative experiments, our approach achieved results comparable to high-complexity models under more limited hardware conditions, This is a promising indication that our model exhibits more efficiency compared to the full attention model.
>
> It should be noted that at the time of writing our paper, we were not aware of the existence of the competitor, Exphormer, which was published at the **ICML conference on July 24th.** We appreciate your feedback on the experimental results. We do recognize the competitiveness of the sparse deformable graph method pointed out by the reviewers.
>
> While both our approach and Sparse Transformers for Graphs share the common goal of reducing the computation complexity of transformer models for graphs, it is important to highlight the differences between the two approaches. Sparse Transformers for Graphs introduces two novel techniques called **virtual global nodes** and **expander graphs**. The virtual global nodes are additional super nodes that connect all nodes in the graph as a global attention storage. The expanders is based on a random learning process without a guaranteed optimal solution. **In contrast, our method incorporates hub labeling technique, which provides a strong theoretical guarantee by transforming global attention into a two-hop attention.** This framework ensures complete information coverage through a two-step process and incorporates a core-fringe structure that requires carefully designed algorithms to ensure the theoretical guarantee of 2-hop coverage.
>
> **In the revised version, we have included the Exphormer as a competing baseline to enhance the credibility and relevance of our experimental results.** Thank you very much for your guidance and suggestions. The table below displays the benchmark results in comparison to Exphormer. **Our technique achieved marginal superiority on 4 benchmarks, whereas Exphormer only won 2**.
>
> |  | PATTERN | CLUSTER | PascalVOC-SP | COCO-SP | Peptides-func | Peptides-struct |
> | --- | --- | --- | --- | --- | --- | --- |
> |  | Accuracy ↑ | Accuracy ↑ | F1 score ↑ | F1 score ↑ | AP ↑ | MAE ↓ |
> | CFGNN | 86.821 ± 0.026 | 78.863 ± 0.032 | 0.3847 ± 0.0273 | 0.2810 ± 0.0095 | 0.6581 ± 0.0047 | 0.2477± 0.0059 |
> | EXPHORMER | 86.74±0.015 | 78.07 ± 0.037 | 0.3975 ± 0.0037 | 0.3455 ± 0.0009 | 0.6527 ± 0.0043 | 0.2481 ± 0.0007 |
>
> ## W2: Motivated by the scalability, but the selected datasets are not large-scale, and the time complexity analysis is not formal.
>
> We selected benchamark datasets such as **OGB [1]** and **LRGB [2]**, which are widely recognized and commonly used in graph learning research. Therefore, we followed their usage to ensure a fair comparison. By using these widely accepted datasets, we aimed to ensure that our comparative analysis aligns with the current practices in the field of graph learning.
>
> For graph tasks, current large-scale graph task datasets primarily focus on the number of graphs rather than the size of individual graphs. The main advantage of our method lies in its **cost-effectiveness and scalability**. Our experimental setup was intentionally designed to operate under compact hardware conditions, utilizing a single RTX3090 with 24GB of memory. Despite these resource constraints, we achieved comparable, and in some cases, superior results compared to existing methods: Graphormer(8x v100 32GB) and GraphGPS(1 × Tesla A100, 40GB).
>
> In addition, we included a comprehensive theoretical analysis of the comparison of computational complexity in **Appendix 7** **(original submission)**.

---

> ### Author Response · Authors · 2023-11-22
> **Response to Reviewer iyTY (2/2)**
>
> ## W3: Section 2.1 is not informative, and its relation with the following proposed method is not clear.
>
> Based on your feedback, we have made revisions to Section 2.1 in order to underline the significance of our work and clarify the distinction between the conventional concept of **receptive field** on graph and our proposed concepts of **information coverage**.
>
> In Section 2.1, the concept of information coverage in graph model learning is initially defined. The motivation behind our approach is to ensure that the embedding of each node on the graph **sufficiently covers all other nodes** on the graph and their associated features. Our main focus is the **structural**, **positional**, and **path information between nodes**. Our contribution mostly lies in proposing the use of hub labeling to construct precise location embeddings. And utilize a more cost-effective approach to achieve broader information coverage, including node features and simple path information coverage, among other aspects. We ensure the model can extract information regarding at least one shortest path for each pair of nodes. Overall, maximizing information coverage is the core value of our work.
>
> ## W4: It is better to add necessary contexts for proposed equations.
>
> We appreciate your feedback and incorporated additional elucidations of the equations in the revised edition. We have revised the latest version to clarify the two notations from Q2 and Q3.
>
> ## W5: It is better to add references for the selected benchmarks and baselines.
>
> The benchmarks and baselines' reference is placed in Appendices 8 and 9 **(original submission)** due to space constraints.
>
> ## Q1: How core nodes are selected in the proposed method?
>
> As stated in **Appendix 3 (original submission)**, we have included a comprehensive example to demonstrate the process of constructing the core-fringe. **The selection of the core nodes is determined by the chosen hub labeling method**. There are numerous methods available for constructing hub labeling. Among those heuristic selection has been proven to be effective in practice [3,4,5]. The criterion is always constructing fewer edges to satisfy the two-hop shortest path coverage. This criterion also motivates our approach: to obtain more information coverage at a lower cost. The approach has been proven effective through extensive experiments with realistic datasets, as demonstrated by [4]. **Hence, we chose to use the method hub pushing with DHP [3] to determine the importance ranking of nodes based on their degree in this wor**k.
>
> ## Q2: In Eq. 3, how the matrix $\mathbb{A}_N$ is constructed?
>
> $\mathbb{A}_N$ is the adjacency matrix of a graph, typically provided by the dataset.
>
> ## Q3: In Eq.4, how the matrix $\mathbb{A}_L$ is constructed
>
> $A_L$ represents the core-fringe labeling relationship provided by the labeling constructed in hub labeling. $A(u,v) = 1$, if $v \in C(u)$.
>
> [1] Open Graph Benchmark: Datasets for Machine Learning on Graphs. NeurIPS 2020
>
> [2] Long Range Graph Benchmark. NeurIPS 2020
>
> [3] Fast Exact Shortest-Path Distance Queries on Large Networks by Pruned Landmark Labeling. Sigmod 2013
>
> [4] An Experimental Study on Hub Labeling based Shortest
> Path Algorithms. VLDB 2017
>
> [5] Hierarchical Hub Labelings for Shortest Paths. ESA 2012